 **eLIFE** elifesciences.org

# The evolution of distributed sensing and collective computation in animal populations

Andrew M Hein[1]*[†], Sara Brin Rosenthal[2,3][†], George I Hagstrom[1][†], Andrew Berdahl[4], Colin J Torney[5], Iain D Couzin[3,6]*

[1]Department of Ecology and Evolutionary Biology, Princeton University, Princeton, United States; [2]Department of Physics, Princeton University, Princeton, United States; [3]Department of Collective Behaviour, Max Planck Institute for Ornithology, Konstanz, Germany; [4]Santa Fe Institute, Santa Fe, United States; [5]Centre for Mathematics and the Environment, University of Exeter, Penryn, United Kingdom; [6]Chair of Biodiversity and Collective Behaviour, University of Konstanz, Konstanz, Germany

**Abstract** Many animal groups exhibit rapid, coordinated collective motion. Yet, the evolutionary forces that cause such collective responses to evolve are poorly understood. Here, we develop analytical methods and evolutionary simulations based on experimental data from schooling fish. We use these methods to investigate how populations evolve within unpredictable, time-varying resource environments. We show that populations evolve toward a distinctive regime in behavioral phenotype space, where small responses of individuals to local environmental cues cause spontaneous changes in the collective state of groups. These changes resemble phase transitions in physical systems. Through these transitions, individuals evolve the emergent capacity to sense and respond to resource gradients (i.e. individuals perceive gradients via social interactions, rather than sensing gradients directly), and to allocate themselves among distinct, distant resource patches. Our results yield new insight into how natural selection, acting on selfish individuals, results in the highly effective collective responses evident in nature.

*For correspondence: ahein@princeton.edu (AMH); icouzin@orn.mpg.de (IDC)

[†]These authors contributed equally to this work

## Introduction

In many highly coordinated animal groups, such as fish schools and bird flocks, the ability of individuals to locate resources and avoid predators depends on the collective behavior of the group. For example, when fish schools are attacked by predators, 'flash expansion' (*Pitcher et al., 1993*) and other coordinated collective motions, made possible above a certain group size, reduce individual risk (*Handegard et al., 2012*). Similarly, fish can track dynamic resource patches far more effectively when they are in a group (*Berdahl et al., 2013*). When an individual responds to a change in the environment (e.g., predator, resource cue), this response propagates swiftly through the group (*Rosenthal et al., 2015*), altering the group's collective motion. How are such rapid, coordinated responses possible? These responses may occur, in part, because the nature of social interactions makes animal groups highly sensitive to small changes in the behavior of individual group members; theoretical (*Couzin et al., 2002*; *D'Orsogna et al., 2006*; *Kolpas et al., 2007*) and empirical (*Tunstrøm et al., 2013*; *Buhl et al., 2006*) studies of collective motion have revealed that minor changes in individual behavior, such as speed (*Tunstrøm et al., 2013*), can cause sudden transitions in group state, reminiscent of similarly sudden phase transitions between collective states in physical systems (such as the solid-liquid-gas transitions as a function of increasing temperature). It has been

 

**eLife digest** In nature, we see many examples of highly coordinated movements of groups of individuals; think of a flock of birds turning swiftly in unison or a crowd of people filing through the exit of a building. A common feature of these behaviors is that they occur without any centralized control, and that they involve sudden and often dramatic changes in the 'collective state' of the group (i.e. speed, or the distances between individuals). In the past, researchers have likened these transitions in collective behavior to phase transitions in physical systems, for example, the transition between liquid water and water vapor. However, it is not clear how such collective responses could have evolved.

Natural selection is an evolutionary process whereby individuals with particularly 'fit' traits produce more offspring than others. Over many generations, these beneficial traits tend to become more common in the population. Hein, Rosenthal, Hagstrom et al. developed a mathematical model to investigate whether the capacity of a population to perform collective motions could evolve through natural selection.

The model shows that over many generations, populations consistently evolve a unique collective trait whereby small responses of individuals to an environmental cue can cause spontaneous changes in the collective state of the local population. These transitions in collective state greatly enhance the ability of individuals to locate and exploit resources. Hein, Rosenthal, Hagstrom et al.'s findings suggest that natural selection acting on the behavior of individuals can cause a population to evolve a distinctive, collective behavior.

The next challenge will be to identify a biological system in which the evolution of collective motion can be studied experimentally to test these predictions.

proposed that individuals may trigger such changes in collective state by responding to the environment, thereby initiating a coordinated response at the group level (e.g., *Couzin et al. (2002)*; *Kolpas et al. (2007)*; *Couzin and Krause, 2003*). This mechanism requires that the behavioral rules of individual animals within a population have evolved in a way that allows groups to transition adaptively among distinct collective states. The evolutionary processes that could lead to this population-level property, however, remain poorly understood.

The feedback between the behavioral phenotypes of individuals, the collective behaviors that these phenotypes produce, and individual-level fitness consequences has made it challenging to study how complex collective behaviors evolve (*Torney et al., 2011*). Many species, including fish and birds, form groups in which members have low genetic relatedness, which implies that kin selection alone cannot explain the evolution of collective behavior. Moreover, while natural selection acts on the behavioral phenotypes of selfish individuals, collective behaviors are group-level, or perhaps even population-level, properties rather than heritable individual phenotypes. To understand how collective behaviors evolve, then, one must first understand the mapping between individual phenotypes and collective behavior, and between collective behavior and individual fitness.

Here, we take advantage of detailed studies of the social interaction rules and environmental response behaviors of schooling fish (*Berdahl et al., 2013*; *Katz et al., 2011*) to develop a biologically-motivated evolutionary model of collective responses to the environment. Using analytical methods and evolutionary simulations, we study how individual behavioral rules produce collective behaviors, and how collective behaviors, in turn, govern the fitness and evolution of selfish individuals. To relate individual and collective behaviors to fitness, we consider a fundamental task faced by fish and other motile organisms: finding and exploiting dynamic resources (*Stephens et al., 2007*). In our model, individuals respond to the locations of near neighbors and also to local measurements of resource quality. Each individual achieves a fitness determined by the resource level it experiences over its lifetime. We use this framework to explore the evolution of complex collective responses to the environment, and how such responses are related to transitions in collective state.

## Model development

### Behavioral rules

We model the movement behaviors of each individual in a population of size $N$ using two experimentally-motivated (*Berdahl et al., 2013*; *Katz et al., 2011*) behavioral rules: a social response rule and an environmental response rule. The social response rule is motivated by experimental studies of pairwise interactions among golden shiners (*Notemigonus crysoleucas*) (*Katz et al., 2011*). Individual fish avoid others with whom they are in very close proximity. As the distance between individuals increases, however, interactions gradually change from repulsive to attractive, with maximum attraction occurring at a distance of two-four body lengths. For longer distances, individuals still attract one another but the strength of attraction decays in magnitude (Appendix section 1; *Katz et al., 2011*). As found in experimental studies of golden shiners (*Katz et al., 2011*) and mosquitofish (*Gambusia holbrooki*) (*Herbert-Read et al., 2011*) there need not be an explicit alignment tendency; rather alignment can be an emergent property of motion combined with the tendencies for repulsion and attraction described above.

To capture these observed social interactions (or 'social forces'), we model the acceleration of individuals using a force-based method (*Katz et al., 2011*). The $i$th individual responds to its neighbors using the following rule:

$$\mathbf{F_{s,i}} = -\nabla \left[ \sum_{\mathbf{j} \in \mathbf{N_i}} \mathbf{C_r} \mathbf{e}^{-|\mathbf{x_i}-\mathbf{x_j}|/l_r} - \mathbf{C_a} \mathbf{e}^{-|\mathbf{x_i}-\mathbf{x_j}|/l_a} \right], \tag{1}$$

where $\mathbf{F_{s,i}}$ is the social force on the $i$th individual, $\mathbf{x_i}$ is the position of the $i$th individual, $\nabla$ is the two-dimensional gradient operator, the term in brackets is a social potential, $C_a$, $C_r$, $l_a$, and $l_r$ are constants that dictate the relative strengths and length scales of social attraction and repulsion, and the set $N_i$ is a set of the $k$ nearest neighbors of the $i$th individual, where a neighbor is an individual within a distance of $l_{max}$ of the focal individual. *Equation 1* does not include explicit alignment with neighbors. A similar model is discussed in *D'Orsogna et al. (2006)*. In *Equation 1*, $l_{max}$ determines the length scale over which individuals are influenced by social interactions. If $l_{max}$ is greater than $l_r$ but less than $l_a$, individuals repel one another at short distances but do not attract one another. We refer to such individuals as asocial (Appendix section 1). If $l_{max}$ is greater than both $l_r$ and $l_a$, individuals repel one another at short distances and are attracted to one another at intermediate distances as observed by *Katz et al. (2011)*. Finite $k$ ensures that individuals can only respond to a limited number of their neighbors in crowded regions of space and provides a simplified model of sensory-based social interactions (e.g., *Rosenthal et al. (2015)*; *Strandburg-Peshkin et al. (2013)*). Finite $k$ also ensures that individuals are limited to finite local density (Appendix section 3).

To model the response of individuals to the environment, we develop an environmental response rule based on experimentally-observed environmental responses of golden shiners (*Berdahl et al., 2013*). In particular, in a dynamic, heterogeneous environment, individual golden shiners respond strongly to local sensory cues by slowing down in favorable regions of the environment, and speeding up in unfavorable regions. In contrast, fish respond only weakly to spatial gradients in environmental quality and instead adjust their headings primarily based on the positions of their near neighbors. Accordingly, we model the $i$th individual's environmental response as a function of the level of an environmental cue (in this case, the level of a resource) at its current position:

$$\mathbf{F_{a,i}} = [\mathbf{\Psi_i}(\mathbf{S}(\mathbf{x_i})) - \eta |\mathbf{v_i}|^2] \frac{\mathbf{v_i}}{|\mathbf{v_i}|}, \tag{2}$$

where $\mathbf{F}_{a,i}$ is the autonomous force the $i$th individual generates by accelerating or decelerating in response to the environment, $\Psi(\cdot)$ is a monotonically decreasing function of the value of an environmental cue, $S(\mathbf{x}_i)$ is the cue value at the $i$th individual's position, $\eta$ is a damping term that limits individuals to a finite speed, and $\mathbf{v}_i$ is the $i$th individual's velocity. In the absence of social interactions, individuals travel at preferred speed $v_i^* = \sqrt{\Psi_i/\eta}$ (for $\Psi_i > 0$). Changes in speed are crucial in the schooling behavior of fish (*Tunstrøm et al., 2013*; *Berdahl et al., 2013*), and as we show below, are also responsible for generating effective collective response in our model. Following the experimental results in *Berdahl et al. (2013)* we assume that individuals do not change their headings in response to the cue. In what follows, we refer to 'cue' and 'resource' interchangeably as we model

the case where the cue is the resource itself (see e.g., *Torney et al. (2009)*; *Hein and McKinley (2012)* for cases where the cue is not a resource).

Combining social and environmental response rules yields two equations that govern each individual's movement (in two dimensions):

$$\frac{d\mathbf{x}_i}{dt} = \mathbf{v}_i, \tag{3}$$

and

$$m\frac{d\mathbf{v}_i}{dt} = \mathbf{F}_{s,i} + \mathbf{F}_{a,i}, \tag{4}$$

where $m$ is mass. *D'Orsogna et al. (2006)* explores the behavior of a similar model with $\Psi_i = \Psi$ constant over the full parameter space. Here we focus on a parameter regime that yields behavioral rules that match the experimental observations of *Katz et al. (2011)* and *Berdahl et al. (2013)*.

We simulate a discretized version of the system described by *Equations 3 and 4*. In particular, we choose a time step, $\tau$, within which the acceleration due to social influences (*Equation 1*) and resource value $S(\mathbf{x}_i)$ are assumed to be constant. Positions, speeds, and accelerations of all individuals at time $t + \tau$ are then given by the solutions to *Equations 3 and 4* at time $t + \tau$, with the values of $S(\mathbf{x}_i)$ and $|\mathbf{x}_i - \mathbf{x}_j|$ determined at time $t$. A navigational noise vector of small magnitude $\gamma$ and uniform heading 0 to $2\pi$ is added to the velocity of each agent at each time step. Taking the limit as $\tau$ goes to zero means that individuals are constantly acquiring information and instantaneously altering their actions in response. In Appendix section 3−6, we analyze a continuum approximation of this limiting model and below we discuss results of this analysis alongside simulation results.

The social interaction rule allows us to build an interaction network for the entire population. Two individuals are socially connected if at least one of them influences the other through *Equation 1*. We define a 'group' as a set of individuals that belong to the same connected component in this network.

## Evolutionary dynamics

The natural environments in which organisms live are often heterogeneous and dynamic (*Stephens et al., 2007*). Consequently, we simulate populations of individuals in dynamic landscapes, where individuals make decisions in response to local sensory cues (local measurements of a resource) and these decisions have fitness consequences for the individuals within the population (*Guttal and Couzin, 2010*; *Torney et al., 2011*). In keeping with experimental observations (*Berdahl et al., 2013*), we assume individuals follow a simple environmental response function: $\Psi_i = \psi_0 - \psi_1 S(\mathbf{x}_i)$, where $\psi_0$ dictates the $i$th individual's preferred speed when the level of the environmental cue is zero and $\psi_1$ determines how sensitive the $i$th individual is to the cue value (*Berdahl et al., 2013*). Rather than prescribing values of $\psi_0$ and $\psi_1$, we use an evolutionary framework similar to that developed by *Guttal and Couzin (2010)* to allow these two behavioral traits to evolve along with the maximum interaction length $l_{max}$, which determines whether individuals are social ($l_{max} >$ length scale of social attraction) or asocial ($l_{max} <$ length scale of social attraction, Appendix section 1).

In each generation, $N$ individuals are located in a two-dimensional environment in which each point in space is associated with a resource value that changes over time (see Materials and methods). Individuals move through the environment using the interaction rules described above, and each individual has its own value of the $\psi_0$, $\psi_1$, and $l_{max}$ parameters. At the end of each generation, we compute each individual's fitness as the mean value of the resource it experienced during that generation. Each individual then reproduces with a probability proportional to its relative fitness within the population. $N$ offspring comprise the next generation where each offspring inherits the traits of its parent modified by a small mutation (Appendix section 2). For reference, we compare the evolution of populations in which $\psi_0$, $\psi_1$, and $l_{max}$ are allowed to evolve, to the evolution of populations of asocial individuals, for which $l_{max}$ is set to a constant (Appendix section 1).

## Results

### Evolution of behavioral rules

In populations of asocial individuals, the baseline speed parameter and environmental sensitivity increase consistently through evolutionary time (*Figure 1A–B*). Asocial individuals move through the environment, slowing down in regions where the resource value is high and speeding up when the resource value is low (*Video 1*). As one would expect from random walk theory (*Schnitzer, 1993*; *Gurarie and Ovaskainen, 2013*), individuals more rapidly encounter regions of the environment with high resource value when they travel at high preferred speeds (*Equation A65*; *Gurarie and Ovaskainen, 2013*), and the more they reduce speed in regions of the environment with high resource quality, the more time they spend in these regions (*Schnitzer, 1993*). Because of these two effects, the fittest asocial individuals have high baseline speeds (i.e., high $\psi_0$) and accelerate and decelerate rapidly in response to changes in the resource value (i.e., high $\psi_1$; *Figure 1A–B*, Appendix).

When populations are allowed to evolve sociality, the evolutionary process selects for very different behaviors (*Figure 1C–E*). Selection quickly favors sociality, and individuals evolve large maximum interaction lengths (*Figure 1C*). Over evolutionary time, selection removes individuals with high and low values of $\psi_0$ and $\psi_1$ from the population and an evolutionarily stable state (ESSt; *Maynard Smith, 1982*) emerges that is characterized by a single mode at the dominant value of each trait (*Figure 1D–E*; Appendix section 2). The ESSt resulting from selection on $\psi_0$, $\psi_1$, and $l_{max}$ is robust in that it is resistant to invasion by phenotypes near the ESSt, and by invaders with trait values far from the ESSt (Appendix section 2). Throughout evolution, populations of social individuals achieve mean fitness values that are approximately five times higher than those of asocial populations, and a coefficient of variation in fitness approximately four times lower than that of asocial individuals (*Figure 1F*).

Notably, a single individual drawn from a population at the ESSt can invade a resident population of asocial individuals and the social strategy quickly sweeps through the population (Appendix section 2). To understand why this invasion occurs, consider a population of asocial individuals that slow down in favorable regions of the environment. If the environment does not change too rapidly, such individuals will accumulate in regions where the resource level is high. This phenomenon has been studied mathematically in the context of position-dependent diffusion (*Schnitzer, 1993*), and will occur, in general, when individuals lower their speeds in response to the value of an environmental cue. A social mutant that responds to the environment, and to its neighbors, can take advantage of the correlation between density and resource quality by climbing the gradient in the density of its neighbors (*Equation 1*). In this case, the positions of neighbors contain information about the value of resources and social mutants quickly invade asocial populations leading to a rapid increase in mean fitness (Appendix section 2).

### Evolved populations collectively compute properties of the environment

The high fitness of the evolved phenotype is due, in part, to a collective resource tracking ability, similar to that found in golden shiners (*Berdahl et al., 2013*). Evolved individuals can find and track resource peaks as they move through the environment (*Figure 2A*, *Video 2*; Materials and methods), whereas asocial individuals and social individuals with trait values far from the ESSt cannot (*Videos 1*, *3–4*). Tracking occurs via a dynamic process. Individuals near the edge of the peak move rapidly, whereas individuals nearer to the peak center (where the resource value is high) move slowly (*Equation 2*). As in fish schools (*Berdahl et al., 2013*), individuals turn toward near neighbors (*Equation 1*) and travel toward the peak center. This collective tracking behavior is particularly important when the resource field changes rapidly over time. As a resource peak moves, individuals at its trailing edge experience a resource value that becomes weaker through time (*Figure 2A*). As the resource value becomes weaker, these individuals accelerate (*Equation 2*), but turn toward neighbors on the peak (*Equation 1*) and thus travel toward the moving peak (*Figure 2A*). When the environment contains multiple resource peaks, evolved populations fuse spontaneously to form groups whose sizes correspond to that of the peak they are tracking (*Figure 2B*), even though no individual is able to assess peak size, or know whether there are multiple peaks in the environment. This

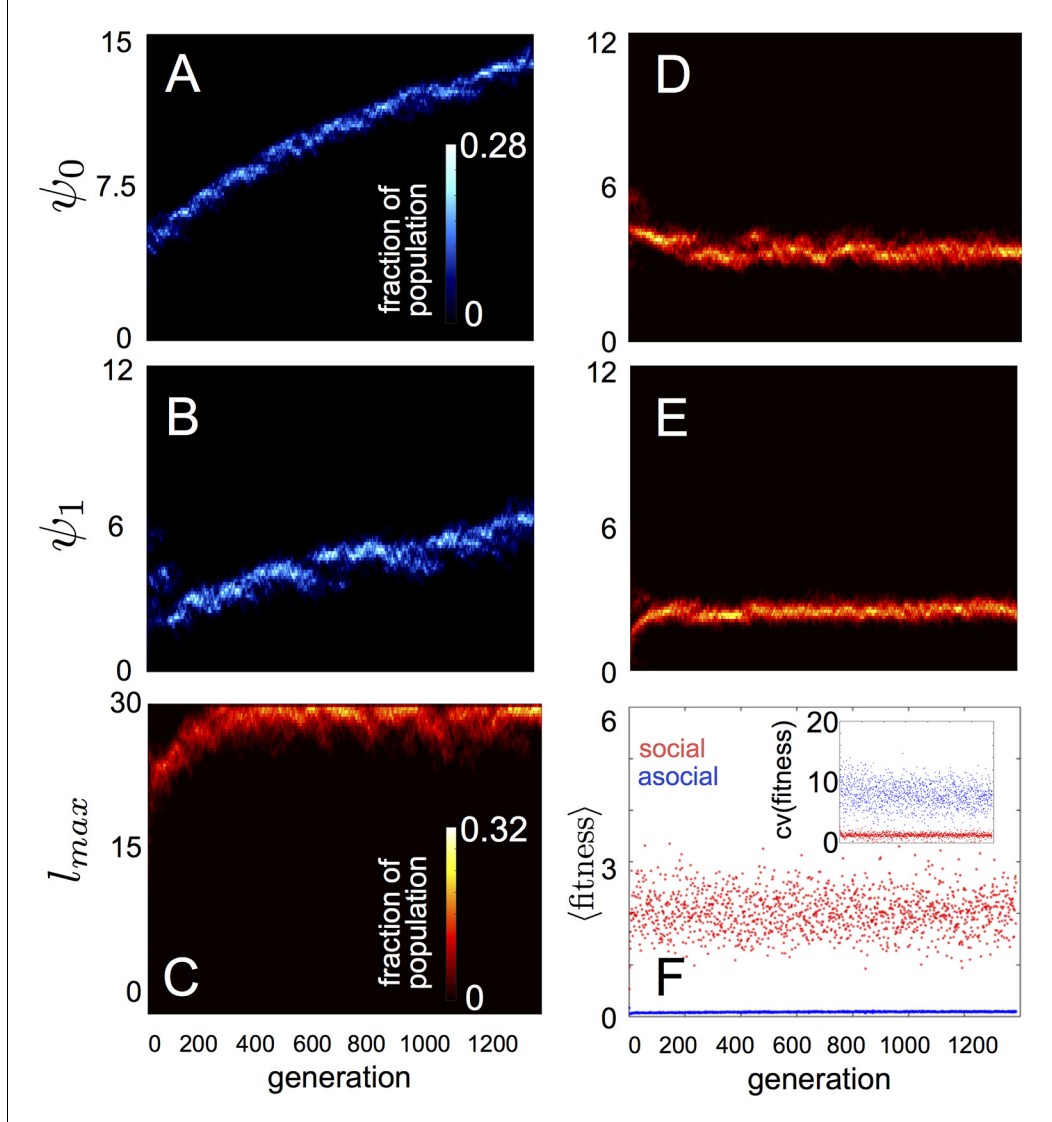

**Figure 1.** Evolution of behavioral rules. (**A, B**) show evolutionary dynamics of populations of asocial individuals (i.e., maximum length scale of social interactions $l_{max}$ fixed; see text). (**C-E**) show evolutionary dynamics of individuals in which the maximum length scale of social interactions $l_{max}$ is allowed to evolve. Brightness of color indicates the frequency of a phenotype in the population. In asocial populations, baseline speed parameter $\psi_0$ (**A**) and environmental sensitivity $\psi_1$ (**B**) increase continually through evolutionary time. When $l_{max}$ is allowed to evolve (**C**), individuals quickly become social ($l_{max}$ approaches maximum allowable value of 30), and baseline speed parameter $\psi_0$ (**D**) and environmental sensitivity $\psi_1$ (**E**) stabilize at intermediate values. Mean fitness of social populations (**F**, red points) is over five times higher than mean fitness of asocial populations (**F**, blue points), and the coefficient of variation in fitness is over four times lower in social populations (**F** inset). Unless otherwise noted, parameter values in all figures are as follows: $C = \frac{C_r}{C_a} = 1.1$, $l = \frac{l_r}{l_a} = 0.13$, $N = 500$, $k = 25$, $\gamma = 0.01$, $\tau = 1$, $m = 1$, $\nu = 1$, $\rho = 0.16$, $M = 2$, $\lambda_0 = 10$, $\lambda_1 = \sqrt{20}$, $\alpha = (1, 0)$, $\beta = 0.1$, and $\tau_p = 1500$.

behavior is consistent with recent sonar observations of foraging marine fish showing that fish form shoals that match the sizes of dynamic resource patches (*Bertrand et al., 2008*; *Bertrand et al., 2014*). Our model demonstrates that collective tracking behavior similar to that observed in real fish schools can evolve through selection on the decision rules of individuals.

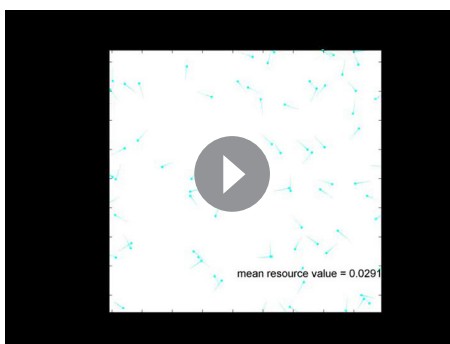

**Video 1.** Asocial population. Responses of population of asocial individuals (points) and dynamic resource peak (resource value shown in grayscale; dark regions have high resource value, light regions have low resource value). Length of tail proportional to speed. Peak centroid moves according to 2D Brownian motion with drift vector $\alpha$ and standard deviation $\beta$ (see Materials and methods). In **Videos 1–4**, view is zoomed in to area surrounding moving resource peak (field of view is $50l_r \times 50l_r$, where $l_r$ is the length scale of repulsion; full environment is projected onto a torus with edge length $346l_r$). Behavioral parameters as follows: $C_r = 1.1$, $C_a = 1$, $l_r = 1$, $l_a = 7.5$, $\gamma = 0.01$, $\tau = 1$, $m = 1$, $\eta = 1$, $\psi_0 = 3$, $\psi_1 = 2.54$. Environmental parameters in **Videos 1–4** are: $\rho = 0.16$, $N = 300$, $M = 2$, $\lambda_0 = 10$, $\lambda_1 = \sqrt{20}$, $\alpha = [0.06 \, 0]$, $\beta = 0.5$.

## Evolved populations are poised near abrupt transitions in collective state

That individuals in evolutionarily stable populations have intermediate baseline speeds and intermediate environmental sensitivities (**Figure 1D–E**) raises a question: what determines the evolutionarily stable values of these traits? It is tempting to conclude that these trait values are determined by the nature of the environment alone. However, the fact that the evolutionary trajectories of social and asocial populations are so different (**Figure 1**), suggests that the collective behaviors discussed above strongly influence the outcome of evolution. Analysis of **Equations 1–4** reveals that the preferred speed parameter divides the dynamical behavior of populations into distinct collective states (**Figure 3**; analysis in Appendix section 5). For $\Psi < 0$, individuals have a preferred speed of zero and the inter-individual distances are governed by initial conditions. In this state, individuals resist acceleration due to social interactions. For small $\Psi > 0$, individuals form relatively dense groups that move through the environment as collectives, either milling, swarming, or translating (**D'Orsogna et al., 2006**), the collective motions exhibited by real schooling fish (**Tunstrøm et al., 2013**). Individual speeds are relatively low and inter-individual distances are short. For large $\Psi$, inter-individual distances are large, and individuals move through the environment quickly. Dynamic changes among theses states are evident in **Video 2**. These collective states are also clearly distinguishable in **Figure 3** ($0 < \Psi < 1.6$ and $\Psi > 2.9$) and **Appendix Figure 9** ($\Psi < 0$), and are separated by abrupt changes in the distances between near neighbors (the inverse of local density, **Figure 3**) or potential energy (**Appendix Figure 9**). The location of transitions between states depends on the parameters of the social response rule (e.g., number of neighbors an individual pays attention to $k$; **Figure 4**). The transitional regimes between these states are reminiscent of the first-order phase transitions that occur in some physical systems, for example at the transition between liquid water and water vapor. As in the liquid-vapor phase transition, transitions in collective state are characterized by strong hysteresis (**Figure 3**). If the population begins with large $\Psi$, mean distance to neighbors remains stable for decreasing $\Psi$ and then decreases abruptly (**Figure 3**, **Appendix Figure 9** upper curve). If $\Psi$ is then increased, mean distance to neighbors increases but follows a different functional relationship with $\Psi$ (**Figure 3**, lower curve). We refer to the collective states as *station-keeping* ($\Psi < 0$; see **Appendix Figure 9**), *cohesive* (small $\Psi$), and *dispersed* (large $\Psi$). The analogy between transitions in collective state in our system and first order phase transitions in physical systems can be made more precise by analyzing the formation rate of groups when $\Psi$ is in the hysteresis region. In the hysteresis region, the rate at which groups of individuals form spontaneously (and therefore nucleate a transition from the *dispersed* to *cohesive* state) depends strongly on $\Psi$; when $\Psi$ is near the upper bound of the hysteresis region, the time required for a group to form spontaneously is very long (see Appendix section 5.4). From a thermodynamic perspective, this makes the spontaneous formation of groups extremely unlikely, which explains why populations that begin in the *dispersed* state follow the upper branch of the hysteresis curve shown in **Figure 3**.

For a wide variety environmental conditions (Appendix section 2) and social parameters (**Figure 4**), the evolutionarily stable trait values have a notable feature: the evolved values of the baseline speed parameter, $\psi_0$, place individuals in the population slightly above the transition between

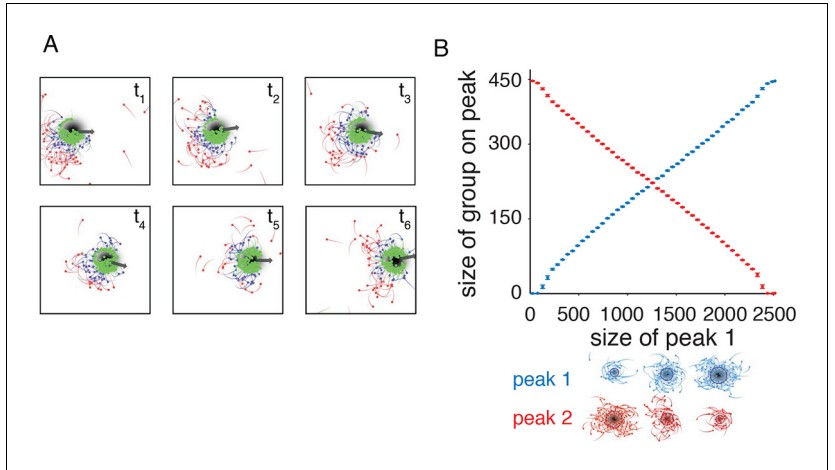

**Figure 2.** Collective tracking of dynamic resource and length-scale matching. (A) Sequence (left to right, top to bottom) of individuals interacting with moving resource peak (resource value in grayscale, darker = higher resource value). Peak is drifting to the right (grey arrow). Colors indicate the regime into which each agent falls (red: $\Psi > 2.95$, blue: $0 < \Psi < 2.95$, green: $\Psi < 0$). Length of tail is proportional to speed. Peak centroid moves according to 2D Brownian motion with drift (see Materials and methods). (B) When environments contain multiple resource peaks, evolved populations divide into groups that match peak sizes, e.g., in a two-peak environment, the size of group on each peak is proportional to peak size. Total size of two peaks is constant so that the larger the first peak (Peak 1, x-axis), the smaller the second peak. Peak size computed as the integral of the resource value over the entire peak (see Materials and methods). Group size is mean size of the group nearest each peak (mean taken over the last 2,500 time steps of each simulation). Points (and error bars) represent mean ($\pm$ 2 standard errors) of 1,000 simulations for each combination of peak sizes. Parameters as in **Figure 1** with $M = 2$ and values of $\psi_0$, $\psi_1$, and $l_{max}$ taken from a population in the ESSt.

*cohesive* and *dispersed* states when $S = 0$ (**Figure 4**, upper panels, **Figure 5**; points in both figures show mean $\psi_0$ values of population in the ESSt), and the evolved environmental sensitivity, $\psi_1$, is large enough that locally, groups of individuals cross from the *dispersed* state through the *cohesive* and *station-keeping* states in regions of the environment where the resource value is high

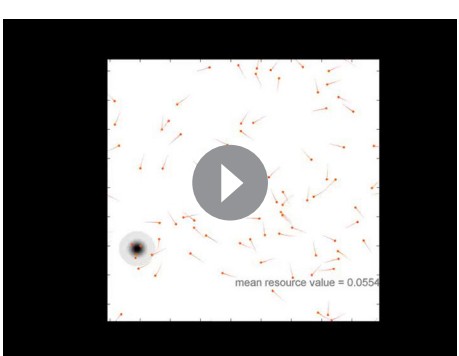

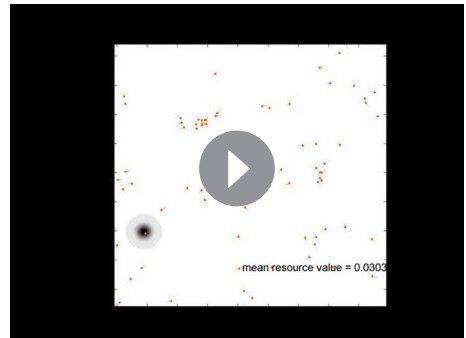

**Video 2.** Population at the evolutionarily stable state (ESSt). Responses of population of individuals evolved for 1500 generations to the ESSt to dynamic resource peaks. Behavioral parameters as in **Video 1** with $k = 25$, $\langle \psi_0 \rangle = 3$, $\langle \psi_1 \rangle = 2.45$, and $\langle l_{max} \rangle = 29$, where $\langle \cdot \rangle$ denotes mean over the population. Note rapid accumulation of individuals near peaks and dynamic peak-tracking behavior of groups.

**Video 3.** Population with mean $\psi_0$ below the ESSt value. Responses of perturbed ESSt population to dynamic resource peaks. All parameters as in **Video 2** except that each individual's value of $\psi_0$ parameter is lowered so that the population mean $\langle \psi_0 \rangle = 0.4$. Note swarms of individuals form in regions of the environment that are far from resource peaks. Individuals explore poorly and therefore have low fitnesses.

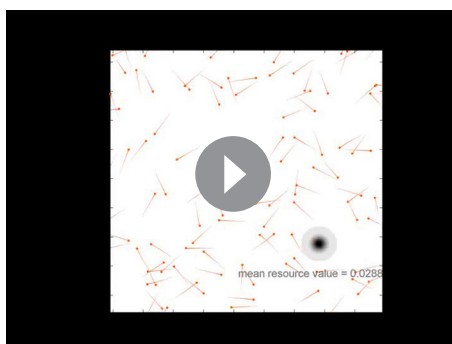

**Video 4.** Population with mean $\psi_0$ above the ESSt value. Responses of perturbed ESSt population to dynamic resource peaks. All parameters as in *Video 2* except that each individual's value of $\psi_0$ parameter is increased so that the population mean $\langle\psi_0\rangle = 8.8$. Note that individuals do not form large groups near resource peaks and fail to track peaks as they move.

(*Figure 2A*, colors indicate instantaneous value of $\Psi$ for each individual). In other words, the evolved values of $\psi_0$ and $\psi_1$ allow local subpopulations to undergo sudden changes from one collective state to another in the proximity of favorable regions of the environment. Importantly, the approximate location of the transition between *cohesive* and *dispersed* states can be predicted by directly analyzing *Equations 1–4* without considering details of the environment, or the mapping between behavior and fitness (*Figure 4* compare upper panels [simulation] to lower panels [analytical prediction]). While the precise evolutionarily stable values of $\psi_1$ depend on the parameters of the environment (Appendix section 2), the evolutionarily stable values of $\psi_0$ place the population near the *cohesive-dispersed* transition in many different kinds of environments (*Appendix Figure 5*). As we show below, being near this transition allows groups to respond quickly to changes in the environment. Our results demonstrate, that such locations in behavioral state-space are, in fact, evolutionary attractors.

The evolutionary results presented in *Figure 1* assume that individuals do not appreciably deplete the resource. We can explore an alternative scenario in which resource peaks are depleted through consumption (Appendix section 2.8). In that case, the $i$th individual consumes resources at a rate $uS(\mathbf{x_i})$ per time step. We repeated evolutionary simulations assuming either a high or low rate of resource consumption $u$. For high consumption rate (100 individuals can deplete a peak in roughly

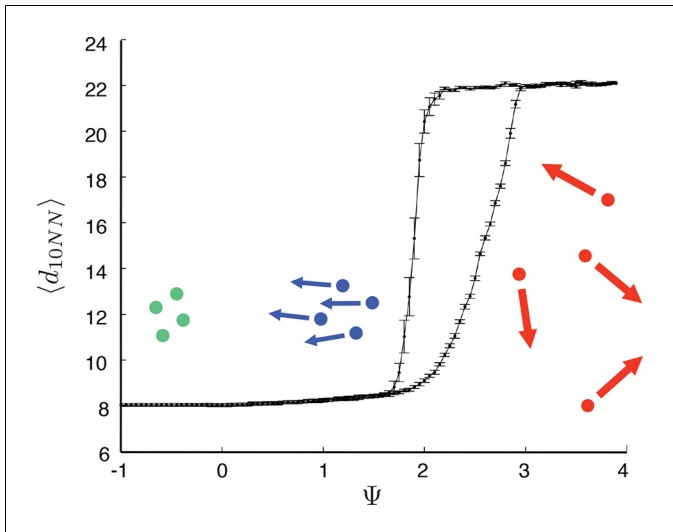

**Figure 3.** Hysteresis plot of the distance to 10 nearest neighbors, averaged over the entire population $\langle d_{10NN}\rangle$ (points and error bars) as a function of preferred speed parameter $\Psi$ in a uniform environment. Figure produced by starting with a population with $\Psi = 4$ in a uniform environment. Population is allowed to equilibrate for 5000 time steps and $\langle d_{10NN}\rangle$ is then computed. $\Psi$ is then lowered. This process is repeated until $\Psi = -1$, at which point the same procedure is used to increase $\Psi$. Upper curve corresponds to decreasing $\Psi$. Lower curve corresponds to increasing $\Psi$. Regimes where $\Psi \sim 0$ and $\Psi \in (1.6, 2.95)$ correspond to transitions between collective states. Points and (error bars) correspond to mean ($\pm$ 2 standard errors) of 50 replicate simulations. Parameters as in *Figure 1* with $l_{\max} = 30$.

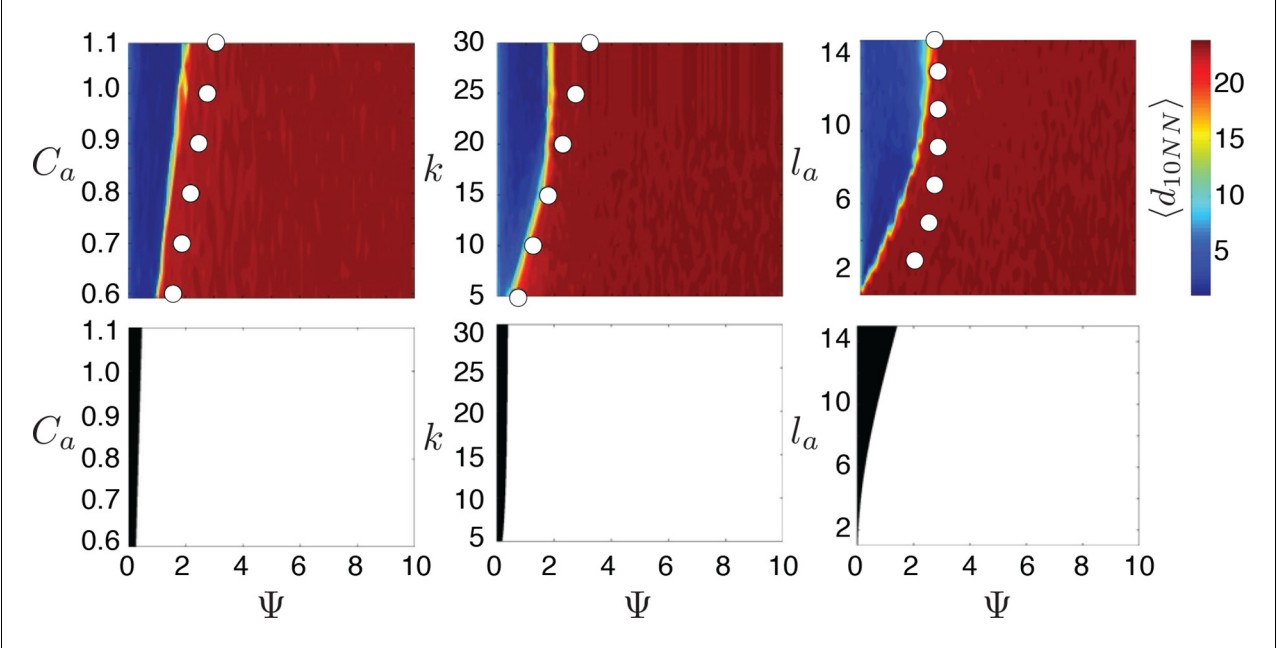

**Figure 4.** Evolved populations are positioned near transitions in collective state. Upper panels show mean distance to 10 nearest neighbors ($\langle d_{10\mathrm{NN}} \rangle$, color scale) from simulated populations. A separate populations is simulated in a uniform environment for each value of the social attraction strength ($C_a$), number of neighbors an individual reacts to ($k$), and the decay length of social attraction ($l_a$) parameters. Red is low density corresponding to *dispersed* state, and blue is high density corresponding to *cohesive* state. Points show the mean value of $\psi_0$ of populations in the EESt (populations evolved for 1,000 generations in an environment with dynamic resource peaks). Evolved populations are positioned near transition between *cohesive* and *dispersed* states. Lower panels are based on analytical calculations and show the predicted regions in which the *dispersed* state is stable (white) and unstable (black, Appendix section 5). Parameters as in **Figure 1** with $M = 15$, $\lambda_0 = 10$, $\lambda_1 = 1.6$, $\alpha = (1, 0)$, $\beta = 0.1$, and $\tau_p = 1500$.

five time steps), $l_{max}$ still increases so that individuals are attracted to one another through social interactions, but selection for large $l_{max}$ is much weaker than the case shown in **Figure 1C** (see **Appendix Figure 7**). Moreover, $\psi_0$ and $\psi_1$ increase continually through evolutionary time. This result is intuitive because when resources are depleted rapidly, the locations of neighbors convey little information about the future location of resources and transitioning from the *dispersed* to *cohesive* state may actually be maladaptive. By contrast, when individuals consume the resource at a more moderate rate (**Appendix Figure 7**), evolutionary trajectories parallel the trajectory shown in **Figure 1C–E**; there is strong selection for high $l_{max}$, $\psi_0$ reaches a stable value that is situated directly above the hysteresis region shown in **Figure 3**, and $\psi_1$ evolves to a stable value that is large enough to allow individuals to cross from *dispersed* to *cohesive*, and *station-keeping* states in regions of the environment where the resource value is high.

## Changes in collective state allow for rapid collective computation of the resource distribution

Why do populations of selfish individuals evolve behavioral rules that place them near the transition between collective states? *Dispersed*, *cohesive*, and *station-keeping* states are each associated with a characteristic density (low, intermediate, and high, respectively; **Figure 3**, **Appendix Figure 9**). If individuals enter the *cohesive* and *station-keeping* states where the resource level is high, the density of individuals becomes strongly correlated with the resource distribution (**Figure 6A**). The similarity between the distribution of individuals and the distribution of the resource can be quantified by the Kullback-Leibler divergence (KL divergence), an information-theoretic concept that measures the distance between two distributions (**Figure 6A** inset). Though individuals cannot sense resource gradients, they can detect gradients in the density of their neighbors (**Equation 1**), and can therefore move up the resource gradient.

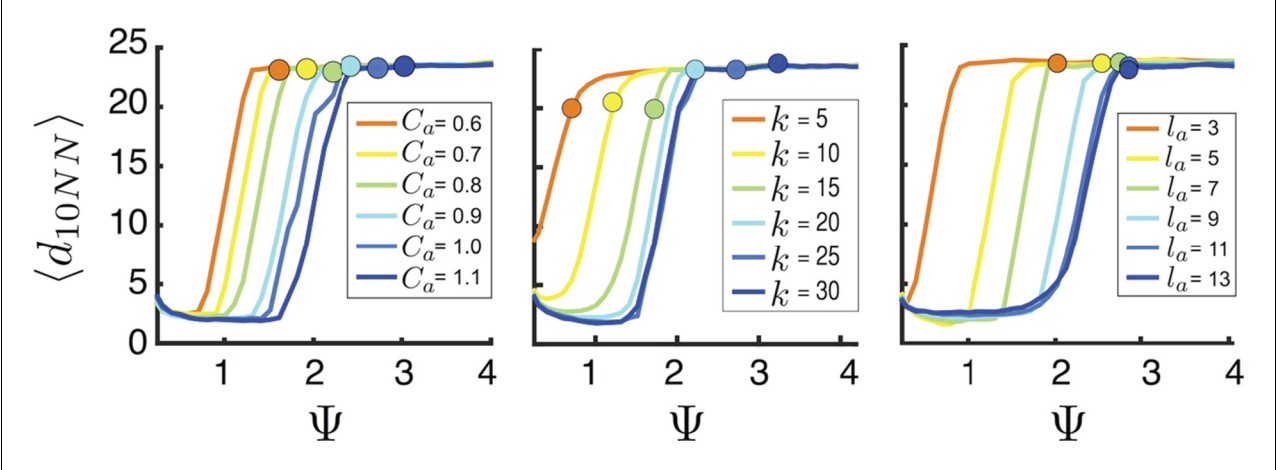

**Figure 5.** Mean distance to nearest neighbors $\langle d_{10\mathrm{NN}} \rangle$ (curves) and ESSt value of $\psi_0$ (points) as a function of social parameters. Points denote mean ESSt value of $\psi_0$. Note abrupt transitions in density as function of $\Psi$, as shown in *Figure 3*. In all cases, ESSt value of $\psi_1$ causes populations to cross transition when resource value is high (i.e., $\psi_0 - \psi_1 \lambda_0 < 0$, where $\lambda_0$ is maximum resource value of each peak). Densities and ESSt values generated as described in *Figure 4*.

The abrupt transitions in the density of individuals between *dispersed* and *cohesive* states (*Figure 3*) mean that there is a strong density gradient in regions of the environment where individuals in the *dispersed* state border individuals in the *cohesive* state (e.g., *Figure 2A*, *6A*, *Video 2*). This suggests that the behavior of an individual in this region can be approximated by considering only its interactions with individuals that are on the resource peak (i.e., where density is high). Using this assumption, we derive analytically the rate at which new individuals join (or rejoin) a group on the resource peak (Appendix section 6.5). Asocial individuals arrive at a resource peak at a rate $\kappa_a$, where $\kappa_a$ is a constant (*Figure 6B*, blue curves and points; *Equation A65*). However, social

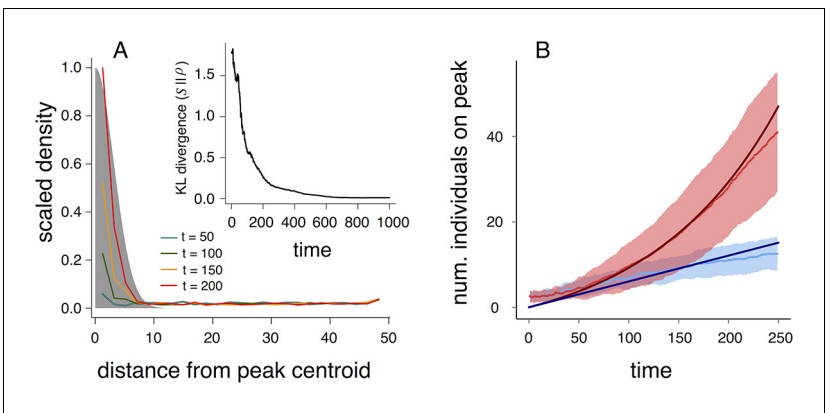

**Figure 6.** Collective computation and social gradient climbing. (**A**) Collective computation of the resource distribution (grayscale represents resource value, normalized to maximum of 1). Curves show local density of individuals at different distances from the resource peak center (maximum value also normalized to 1). Note the rapid accumulation of individuals near the peak center. The distribution of individuals becomes increasingly concentrated in the region where the resource level is highest; inset shows that the Kullback-Leibler divergence between the resource distribution and the local density of individuals decreases through time as the two distributions become more similar. (**B**) Number of individuals near peak center (within one decay length, $\lambda_1$, of peak center) as a function of time. Red and blue points and confidence bands represent means $\pm 1$ sd. for 100 replicate simulations. Red points and band is ESSt population and blue points and band is an asocial population with the same parameter values. Curves are analytical predictions based on *Equations 3 and 4* (Appendix section 6).

individuals initially arrive at a rate that increases as more individuals reach the peak, such that the number of individuals on the peak, $N_s$, increases exponentially with time: $N_s \approx \kappa_{s,1} + \exp(\kappa_{s,2}t)$, where $\kappa_{s,1}$ and $\kappa_{s,2}$ are positive constants (*Figure 6B*, red curves and points; *Equation A68–A70*). Analytical calculations (*Figure 6B*, solid lines) agree well with results of numerical simulations (*Figure 6B*, points and confidence bands). The rapid accumulation shown in *Figure 6* is especially important when the environment changes quickly with time; it allows groups to respond swiftly to changes in the resource field and enables the emergent resource tracking behavior described above.

The form of *Equations (3–4)* implies that an individual's behavioral response combines personal information about the environment (*Equation 2*) with social cues (*Equation 1*). In fact, under a time rescaling, our model is equivalent to one in which the relative strength of social forces varies across the environment (Appendix section 4). The tradeoff between using social information and personal information is inherent in social decision-making (*Couzin et al., 2005*; *Couzin, et al., 2011*). This tradeoff means that individuals with large $\psi_0$ and $\psi_1$ are, by default, less responsive to their neighbors. Perturbing the values of $\psi_0$ and $\psi_1$ of individuals in populations at the ESSt show that, in populations with high mean $\psi_0$, individuals fail to form large groups and are poor at tracking resource peaks (Appendix section 2.6, *Appendix Figure 6*). In populations with high mean values of $\psi_1$, individuals form groups (Appendix section 2.7), but fail to exploit regions with the highest resource quality. Individuals with low values of $\psi_0$ or $\psi_1$ form groups but do not effectively track dynamic resources (Appendix section 2.7).

## Discussion

Our model demonstrates that selection on the behavioral phenotypes of selfish individuals can lead to the rapid evolution of distributed sensing and collective computation. The mechanism that promotes this evolution involves the use of public information: when individuals respond to the environment by slowing down in regions of high resource quality – a behavior that is adaptive even in the absence of social interactions (*Appendix Figure 2*) – their positions become correlated with the locations of resources. Social individuals can exploit this public information by climbing gradients in the density of their neighbors. As in simple, game-theoretic models of social foraging (e.g., *Clark and Mangel, 1984*), social individuals gain a fitness advantage by using information about the environment gleaned by observing neighbors. Because of this, asocial populations are readily invaded by social mutants and collective behaviors evolve (Appendix section 2).

Evolutionarily stable populations occupy a distinctive location in behavioral state space: one in which small changes in individual behavior cause large changes in collective state (*Figures 4*, *5*). When individuals respond to local environmental cues by accelerating or decelerating, local populations transition between the collective states shown in *Figure 3* (e.g. *Figure 2A*). This creates the strong spatial gradient in population density (*Figure 6A*) and allows groups to track dynamic features in the environment rapidly. Perturbations of this evolutionarily stable state cause individuals either to weigh social information too heavily (i.e., small $\psi_0$ and/or $\psi_1$), in which case groups fail to explore effectively (*Video 3*, *Appendix Figure 7*), or to weigh personal information too heavily (i.e., large $\psi_0$ and/or $\psi_1$), in which case individuals fail to exploit the social information that enables dynamic resource tracking (*Video 4*, *Appendix Figure 7*). Because of this, mutants with phenotypes far from the evolutionarily stable state are removed from the population by natural selection. The transitions we observe in collective state bear a resemblance to phase transitions in physical systems, and our results lend credence to the hypothesis that natural selection can result in the evolution of biological systems that are poised near such bifurcation points in parameter space. Importantly, we show that these high-fitness regions of parameter space can be predicted a priori from the structure of individual decision rules, even without knowledge of the environment.

Collective computation is a notion that has strongly motivated research on animal groups (*Berdahl et al., 2013*; *Couzin, 2007*; *Cvikel, et al., 2015*). In our model, populations perform a collective computation through their social and environmental response rules. When individuals are exposed to a heterogeneous resource environment, their responses to the environment cause a modification of the local population density; individuals aggregate in regions where the resource cue is strong. The population performs a physical computation in the formal sense (*Schnitzer, 2002*): physical variables – the positions and relative densities of neighbors – represent mathematical ones – spatially resolved estimates of the quality of resources in the environment. The environments

considered in our study bear a strong resemblance to those encountered in dynamic coverage problems in distributed control theory (*Bachmayer and Leonard, 2002*), dynamic optimization problems (*Passino, 2002*), and Monte Carlo parameter estimation (*McKay, 2003*). Combining an evolutionary approach to algorithm design with collective interactions may therefore be a useful starting point for optimization schemes or control algorithms for autonomous vehicles, particularly if the structure of social interactions leads to bifurcation points in behavioral parameter space as in the model studied here.

Understanding the feedback loop between individual behavior, collective behavior of populations, and selection on individual fitness is a major challenge in evolutionary theory (*Guttal and Couzin, 2010*; *Torney et al., 2011*; *Pruitt and Goodnight, 2014*). Our framework closes this loop and demonstrates how distributed sensing and collective computation can evolve through natural selection on the decision rules of selfish individuals.

## Materials and methods

### Resource environment

Our model of the resource environment incorporates three salient features of the resource environments that schooling fish and other social foragers encounter in nature. These features are: 1) spatial variation in resource quality, 2) temporal variation in resource quality, and 3) characteristic length scales of resource patches (*Stephens et al., 2007*; *Bertrand et al., 2008*; *Bertrand et al., 2014*). Accordingly, we model a two-dimensional environment in which the resource is distributed as a set of $M$ resource peaks. We assume the boundary of the environment is periodic such that individuals, inter-individual potentials, and resource peaks are all projected onto a torus. Each of the $M$ peaks decays like a Gaussian with increasing distance to the peak center. The value of the resource in a single peak at a location, $\mathbf{x_i}$, is given by

$$S(\mathbf{x},\mathbf{x}_s) = \lambda_0 e^{-\frac{|\mathbf{x}-\mathbf{x}_s|^2}{\lambda_1^2}},\tag{5}$$

where $\lambda_0$ is a constant that determines the resource value at the peak center and $\lambda_1$ is a decay length parameter, and $\mathbf{x_s}$ is the location of the centroid of the peak of interest. The total resource value the $i$th individual experiences $S(\mathbf{x}_i)$ is the sum over all peaks in the environment. Each peak moves according to Brownian motion with drift vector $\alpha$ and standard deviation $\beta$. At each time step, each peak has a probability $1/\tau_p$ of disappearing and reappearing at a new location, chosen at random from all locations in the environment.

## Acknowledgements

This work was partially supported by National Science Foundation (NSF) Grants PHY-0848755, IOS-1355061, and EAGER IOS-1251585; Office of Naval Research Grants N00014-09-1-1074 and N00014-14-1-0635; Army Research Office Grants W911NG-11-1-0385 and W911NF-14-1-0431; Human Frontier Science Program Grant RGP0065/2012 (to I.D.C.), NSF Dimensions of Biodiversity grant OCE-1046001, and a James S McDonnell Foundation Fellowship (to A.M.H.).

## Additional information

### Competing interests

IDC: Reviewing editor, *eLife*. The other authors declare that no competing interests exist.

### Funding

| Funder | Grant reference number | Author |
| --- | --- | --- |
| James S. McDonnell Foundation | | Andrew M Hein |
| National Science Foundation | PHY-0848755, IOS-1355061, and EAGER IOS-1251585 | Iain D Couzin |

| Army Research Office | W911NG-11-1-0385 and W911NF-14-1-0431 | Iain D Couzin |
| Office of Naval Research Global | N00014-09-1-1074 and N00014-14-1-0635 | Iain D Couzin |
| Human Frontier Science Program | RGP0065/2012 | Iain D Couzin |
| National Science Foundation | Dimensions of Biodiversity OCE-1046001 | George I Hagstrom |

The funders had no role in study design, data collection and interpretation, or the decision to submit the work for publication.

#### Author contributions

AMH, SBR, GIH, Conception and design, Acquisition of data, Analysis and interpretation of data, Drafting or revising the article; AB, CJT, IDC, Conception and design, Drafting or revising the article

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

## Appendix

# 1 Social interaction rules

## 1.1 Model of social interactions

Past individual-based models that include social interactions have often depicted social interactions by assuming that individuals monitor metric 'zones'. Individuals avoid neighbors in a small zone of avoidance, and align and move toward neighbors within larger zones of social interactions (e.g., *Guttal and Couzin, 2010*; *Couzin et al., 2002*; *Chou et al., 2012*). Here, we use an alternative model that depicts social interactions as forces that act to modify individuals' accelerations. This approach is closely related to force matching methods that have been applied to data to infer the strength of pairwise social interactions among individuals. We assume that social forces depend on distance in a way that creates short-range repulsion among individuals, strong intermediate range attraction, and weak attraction for longer ranges in agreement with results of *Katz et al. (2011)*. We model the social forces on a focal individual, $i$, by the following equation:

$$\mathbf{F}_{s,i} = -\nabla\left[\sum_{j\in N_i} C_r e^{-|\mathbf{x}_i-\mathbf{x}_j|/l_r} - C_a e^{-|\mathbf{x}_i-\mathbf{x}_j|/l_a}\right], \tag{A1}$$

where, as described in the Main Text, $\mathbf{x_i}$ and $\mathbf{v_i}$ are the position and velocity of the $i$th individual, respectively, $\nabla$ is the two-dimensional gradient operator, the term in brackets is a social potential, $C_a$, $C_r$, $l_a$, and $l_r$ are constants, and the set $N_i$ is a set of the $k$ nearest neighbors of the $i$th individual, where a neighbor is an individual within a distance of $l_{max}$ of the focal individual. *Appendix Figure 1* shows the effective force exerted on a focal individual by a neighbor located along the focal individual's trajectory, either behind ($x$-axis $< 0$) or in front of ($x$-axis $> 0$) the focal individual [compare to *Appendix Figure 2* of *Katz et al. (2011)*]. Unlike many past models of interactions among individuals, we do not assume that individuals explicitly align with one another. However, because the r.h.s of *Equation A1* is proportional to the gradient of a social potential, social interactions can cause the focal individual to turn. This turning toward neighbors causes the social gradient climbing behavior described in the Main Text and discussed in detail in Appendix section 6 below.

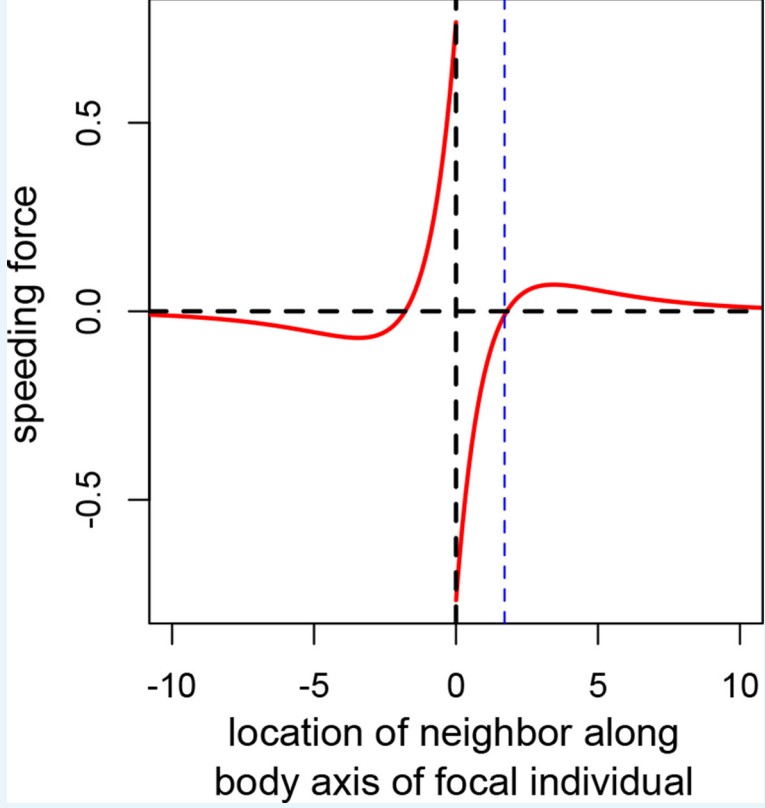

**Appendix Figure 1.** Speeding force as a function of the location of a single neighbor. On the left of the origin, the neighbor is behind the focal individual. For very short distances, the neighbor exerts a positive speeding force on the focal individual, causing it to accelerate. For longer negative distances, the speeding force on the focal individual is negative; the focal individual decelerates to come closer to its neighbor. To the right of the origin the neighbor is in front of the focal individual. For short distances, the speeding force on the focal individual is negative, causing it to slow down. For longer positive distances the speeding force exerted on the focal individual is positive; the focal individual speeds up, closing the distance between it and its neighbor. The distance corresponding to repulsion only is shown with the blue line. Parameters as follows: $C_a = 1$, $C_r = 1.1$, $l_a = 3$, $l_r = 1$.

## 1.2 Definition of an asocial individual

To illustrate the collective behavior and evolution of social individuals it is useful to compare social individuals to individuals that are not influenced by social attraction. We refer to such individuals as 'asocial' and define them in terms of **Equation A1** by setting $l_{max}$ to a value that corresponded to the distance at which the gradient of the social potential for a pairwise interaction is equal to zero (blue line in **Appendix Figure 1**: point at which potential crosses zero). We define asocial agents in this way because the short-range repulsion included in the inter-agent potential shown in **Appendix Figure 1** represents collision-avoidance–a behavior that should be common to all individuals, regardless of whether they are socially attracted to one another. While this definition of an 'asocial' individual is more biologically sensible, we have also tried modeling asocial individuals by assuming that the r.h.s. of **Equation A1** is equal to zero for all individuals (this assumes, for instance, that these individuals are not limited to finite local density); this approach does not qualitatively change the results presented below and in the Main Text.

# 2 Evolutionary dynamics

## 2.1 Selection algorithm

To understand the connection between the evolution of collective behaviors and selection on the performance of individuals, we implement a simple evolutionary algorithm similar to that used in *Guttal and Couzin (2010)*. In the first generation, $N$ agents with heterogeneous values of $\psi_0$ and $\psi_1$ and $l_{max}$ are initiated in an environment with $M$ resource peaks. The number of agents remains constant across generations, and generations are non-overlapping. Each generation consists of a simulation run for 5,000 or 10,000 time steps over which we calculated the mean resource value experienced by each individual in the population. At the end of each generation, $N$ individuals are selected from the population (with replacement) to reproduce themselves, yielding a total of $N$ new offspring. An individual's probability of being selected for reproduction is proportional to its mean resource value, normalized by sum of mean resource value over all individuals in the population. Individuals that perform well are more likely to be selected to reproduce and are likely to produce more offspring than individuals that perform poorly. The selection probability of the $i$th individual $p_i$ is defined as follows:

$$p_i = \frac{\langle S_i(t) \rangle_t}{\sum_{j=1}^{N} \langle S_i(t) \rangle_t}, \tag{A2}$$

where $S_i(t)$ is the instantaneous resource value of individual $i$, and angular brackets represent time-averaging over the particular generation under consideration. If an individual is selected for reproduction, a child is produced in the next generation with $\psi_0$ equal to that of the parent, with a small mutation. The $\psi_0$ value of an offspring is equal to the $\psi_0$ value of its parent, plus a normally distributed random number $\sigma$ with mean zero and variance $\gamma_m$:

$$\psi_{0,i'} = \psi_{0,i} + \sigma, \tag{A3}$$

where $\psi_{0,i'}$ is the $\psi_0$ value of an offspring of individual $i$. The $\psi_1$ and $l_{max}$ traits of offspring were determined in the same way.

## 2.2 Evolution of asocial populations

In general, populations of asocial individuals evolve to have increasing $\psi_0$ and $\psi_1$ values. While fitnesses of individuals in these populations are well below fitnesses of individuals in the evolutionarily stable states discussed below (see Main Text *Figure 1F*), selection on asocial populations still leads to an increase in mean fitness (*Appendix Figure 2*). This occurs because, as evolution progresses and $\psi_0$ and $\psi_1$ values evolve, asocial individuals spend more time in regions of the environment with high resource value.

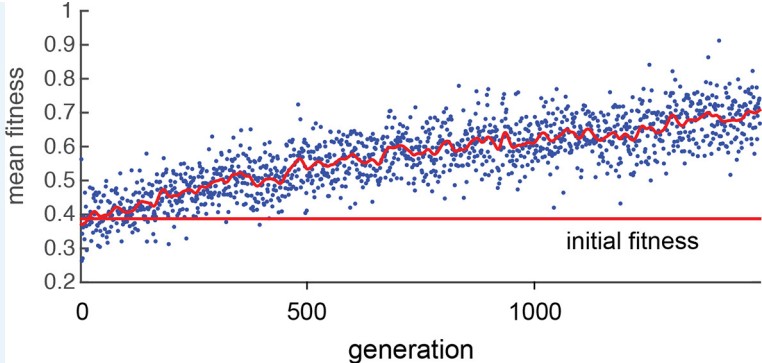

**Appendix Figure 2.** Fitness of asocial population through evolutionary time.
Blue points indicate mean fitness of population in each generation. Horizontal red line
indicates mean fitness of population over first ten generations. Corresponding $\psi_0$ and $\psi_1$
values for each generation are shown in *Figure 1A,B* of the Main Text.

## 2.3 Establishment of evolutionarily stable state (ESSt)

We allow populations to evolve according to the algorithm described above. Initial values of $\psi_0$
and $\psi_1$ phenotypes are drawn at random from uniform distributions between 0 and 6. Initial
values of $l_{max}$ are drawn with uniform probability from the interval $(0, 30)$. The distribution of
trait values quickly stabilizes for all three phenotypic traits as shown in *Figure 1C–E* of Main
Text. We refer to this evolved state as an evolutionarily stable state (ESSt, *Guttal and Couzin,
2010*). The persistent variance in the distribution of $\psi_0$, $\psi_1$, and $l_{max}$ are partially due to
mutations in the value of these traits, which are continually introduced into the population. We
therefore expect such persistent of inter-individual variation in phenotype as a result of
mutation-selection balance.

## 2.4 Robustness of ESSt

To evaluate the robustness of the evolutionarily stable state (ESSt) described in the Main Text,
we performed evolution under invasion by phenotypes that are both near to, and far from the
ESSt. We initiated the population with trait distributions from the ESSt (selected from the final
generation of simulations used to establish ESSt). Then in each generation, we selected
individuals to reproduce and applied ordinary mutations as described above. However, before
initiating the next generation, a single individual was chosen to serve as an invader. That
individual's phenotype was replaced by values ($\psi_0^*$, $\psi_1^*$, and $l_{max}^*$). $\psi_0^*$ and $\psi_1^*$ were chosen with
uniform probability from the interval $(0, 40)$ and $l_{max}^*$ is chosen with uniform probability from
the interval $l_{max} \in (0, 30)$. Though these intervals are somewhat arbitrary, we note that $\psi_0$ and
$l_{max}$ must ultimately be bounded above by limits on the speed that individuals can sustain, and
by limits on the distance over which individuals can perceive one another, respectively. $\psi_1$
should also be bounded above because it is limited by the rate at which individuals can
accelerate (decelerate) in response to changes in the measured value of an environmental cue.
Thus, all three traits are bounded above due to physical constraints. Applying higher bounds
on these trait values did not qualitatively change our conclusions.

*Appendix Figure 3* shows a typical evolutionary progression when a population at ESSt
acquires mutations (i.e., small changes in phenotype) and receives an invader in each
generation. Although invaders from across the phenotype space are introduced into the
population (*Appendix Figure 3* blue dots across phenotype space), none of these invaders
establishes for more than a few generations (*Appendix Figure 3* blue dots become extinct
after few generations). The ESSt is resistant to invasion by both nearby phenotypes,
introduced through ordinary mutation, and phenotypes far from the ESSt, introduced through
invaders. We therefore refer to the ESSt as robust.

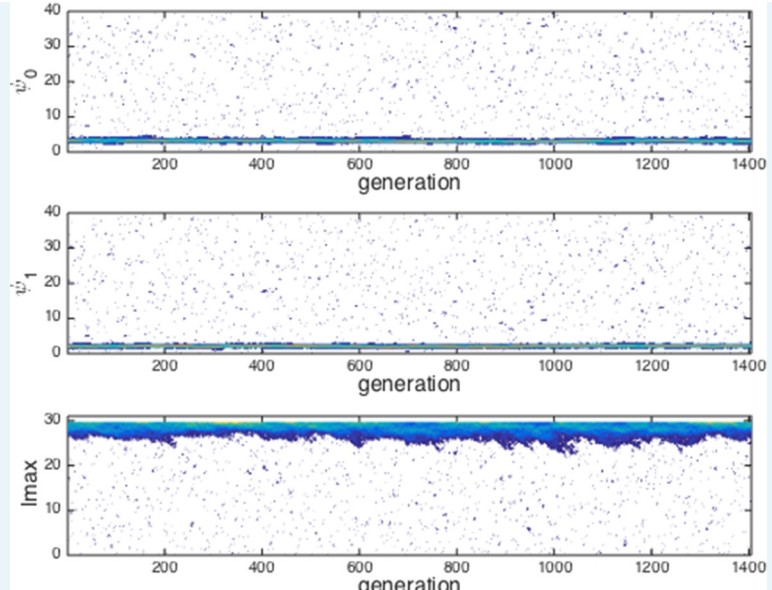

**Appendix Figure 3.** Evolution of traits under invasion by mutants far from the ESSt. Example evolutionary progression for $\psi_0$, $\psi_1$ and $l_{max}$. Note that invaders (blue points introduced across phenotype space) do not establish and the dominant trait values in the population do not change over evolutionary time. Color indicates frequency of phenotype in population (white = 0, blue = low frequency, orange = high frequency).

## 2.5 Invasion of asocial population by social strategy

To determine whether phenotypes from the ESSt could invade a population of purely asocial individuals, we performed another set of evolutionary simulations in which we initiated populations with $N - 1$ asocial individuals and a single individual, chosen at random from the ESSt. *Appendix Figure 4* shows evolutionary progressions from this initial state. In panel A, the full trait distribution of the population is shown. The social invader increases in frequency and sweeps the population of asocials. Replicate invasions show a very similar progression (*Appendix Figure 4B*). The final distribution of trait values matches the ESSt. The change in phenotypes that occur when the ESSt phenotype invades the asocial population lead to a dramatic increase in mean fitness (*Appendix Figure 4C*) and a decrease in the range of fitnesses of different individuals in the population.

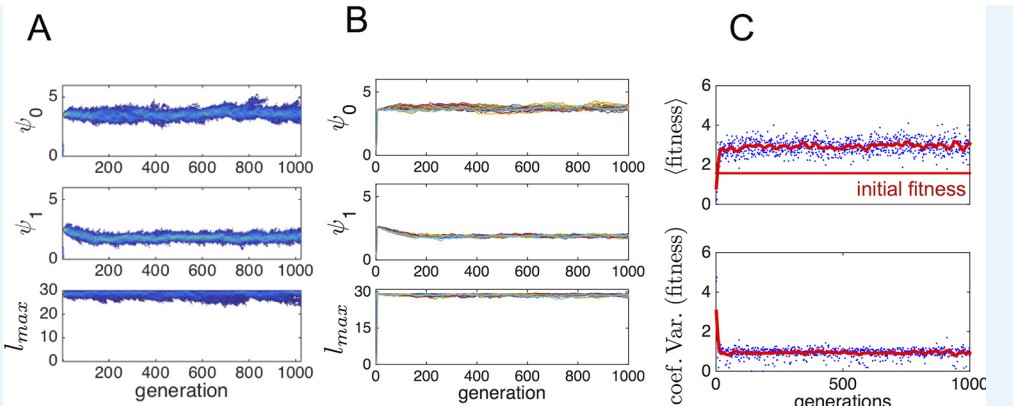

**Appendix Figure 4.** Typical progression of evolution from initial state with N − 1 asocial individuals and individual selected from the ESSt population.
(**A**) Invader from ESSt increases in abundance and sweeps population. (**B**) Mean trait values in 50 replicate invasions (each line is a separate invasion from the same initial state with N − 1 asocial individuals). ESSt phenotype quickly invades and sweeps the population in all cases. (**C**) Mean and coeffient of variation in fitness corresponding to the invasion shown in **A**.

## 2.6 Dependence of evolutionary outcomes on the environment

One of the conclusions drawn in the Main Text is that the trait values of the evolved population at the EESt correspond to a location in behavioral state space where the population is in the dispersed state in regions of the environment with low resource quality, and that the population transitions from the dispersed to cohesive and station-keeping states in regions of the environment with high resource quality. We evaluated whether this conclusion holds, more generally, by evolving populations in more complex environments in which environmental properties were selected at random. We initialized trait values of populations as described in *Establishment of evolutionarily stable state (ESSt)* above. However, to generate the environment, we chose the number of Gaussian resource peaks at random from 1 to 50 with uniform probability. The maximum resource value of each peak and the variance of the two-dimensional Gaussian peak shape were also chosen at random. Maximum resource value was chosen with uniform probability from the interval $(0, 10]$ and variance was chosen with uniform probability from the interval $(0, 30]$. Finally, the variances of all peaks in a given simulation were rescaled so that the sum of the integral of all peaks over the environment was equal to $400\pi$. We enforce this latter condition to ensure that resource peaks are small relative to the size of the environment. All other parameter values were those listed in **Figure 1** of the Main Text, except that $N$ was 300.

We allowed populations to evolve for 1500 generations and recorded values of $\psi_0$, $\psi_1$, and $l_{max}$ that evolved. **Appendix Figure 5** show mean $\psi_0$ and $\psi_1$ trait values after 1500 generations for evolutionary simulations with different environmental conditions. The gray band in **Appendix Figure 5** corresponds to the region of hysteresis between cohesive and dispersed states shown in **Figure 3** in the Main Text. With the exception of a small number of simulated evolutions (**Appendix Figure 5**, points below gray band) populations in all environments had mean trait values of $\psi_0$ in or above the hysteresis region. In all cases, the combination of $\psi_1$ and $\psi_0$ caused individuals to exhibit values of $\Psi = \psi_1 - \psi_0 S(\mathbf{x}_i)$ that were less than zero in the most favorable regions of the environment. Thus, for the large majority of random environmental conditions we generated, individuals transition from $\Psi$ values that correspond to the dispersed state, through $\Psi$ values that correspond to cohesive and station keeping states in favorable regions of the environment.

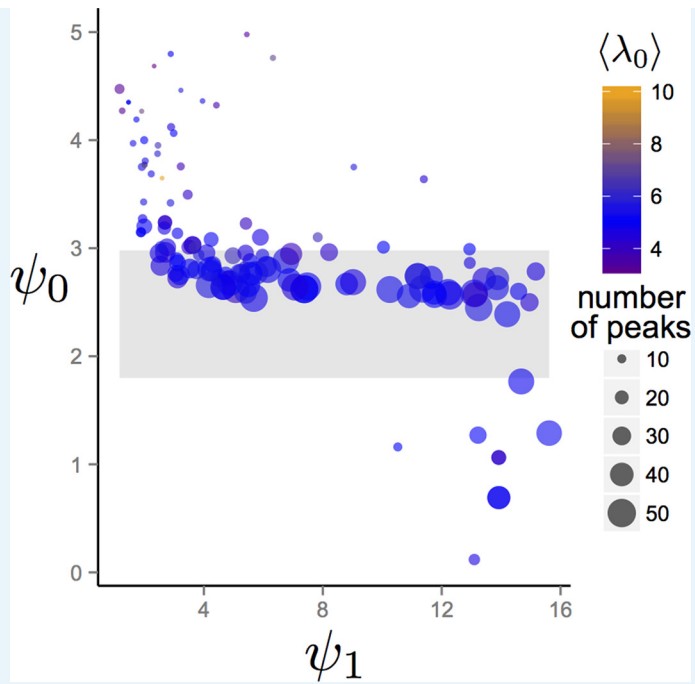

**Appendix Figure 5.** Trait values after 1500 generations of evolution in randomly generated environments.

Each point represents the mean trait values of a single population that has been allowed to evolve for 1500 generations. Point sizes denote the number of peaks that were present in the environment. Point colors represent the maximum resource value $\lambda_0$ averaged over all peaks present in the environment. Gray region corresponds to the region of hysteresis shown in *Figure 3* of the Main Text. Number of peaks and peak parameters were chosen at random. All other parameters as in *Figure 1* of Main Text.

## 2.7 Perturbation of populations around the ESSt

To further understand how the evolutionarily stable trait values lead to high individual fitness we perturbed the entire populations at ESSt by shifting either $\psi_0$ or $\psi_1$ of all individuals in the population. This resulted in a change in the mean value of these traits over the entire population. We then simulated the dynamic behaviors of the new perturbed population in a simplified environment containing two resource peaks. Initially, all individuals were located in a single group near one of the peaks (the starting peak). *Appendix Figure 6* shows that, for fixed $\psi_1$, group sizes and mean fitness vary strongly as a function of the mean value of $\psi_0$ of the population (both $\psi_0$ and $\psi_1$ are taken from a population at ESSt and values of $\psi_0$ are shifted to change $\langle\psi_0\rangle$ of the population). For small values of $\psi_0$, individuals track the starting peak but do not find the second peak (*Appendix Figure 6A*, blue and red points, respectively). As $\psi_0$ reaches approximately 2.2, individuals begin to form a group on the second resource peak (*Appendix Figure 6A*, red points denoting size of group nearest second peak begin to increase). Mean performance of individuals in the group nearest the second peak rapidly increases (*Appendix Figure 6A*, red points rapidly increase for $\psi_0 > 2.2$). When performance is averaged over the entire population, there is a clear maximum at $\psi_0 \sim 3.6$ (*Appendix Figure 6C*), the value corresponding to mean (and modal) $\psi_0$ for the evolved population in the ESSt (orange point in *Appendix Figure 6C*). Selection on fitnesses of individuals and optimization for maximum fitness of the entire population lead to the same value of $\psi_0$. For larger values of $\psi_0$, the average performance over all individuals begins to decline (*Appendix Figure 6C*) because fewer individuals aggregate near peaks. Perturbations of $\psi_1$ also lead do decreases in mean fitness at the population level (*Appendix Figure 6F*). For mean $\psi_1$ below that of the ESSt, individuals form small groups near peaks (*Appendix Figure*

*6D*). For mean $\psi_1$ above that of ESSt, individuals form large groups, but individuals in the groups near peaks have low fitness because they do not effectively aggregate near peak centers (*Appendix Figure 6E*).

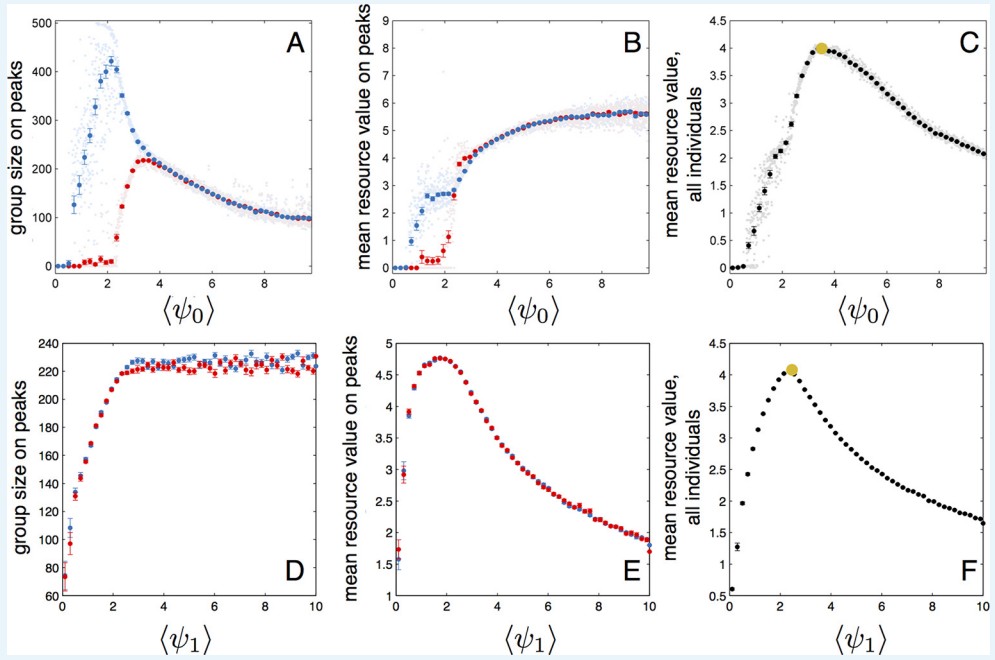

**Appendix Figure 6.** Performance of populations near the evolutionarily stable state. (**A**) The number of individuals on each peak (the starting peak, blue; the second peak, red) as a function of the mean baseline speed parameter, $\psi_0$ of a population perturbed from the ESSt. Below a $\psi_0$ of roughly 2.2, individuals do not form a large group on the second peak. (**B**) Mean resource value of individuals on the starting peak (blue) and second peak (red). (**C**) Resource value averaged over all individuals in the population (individuals in groups nearest each peak and all other individuals in the environment). Note maximum value occurs in the regime where individuals aggregate on both the starting and second peaks ($\psi_0 \sim 3.6$). Orange point indicates values corresponding to ESSt. (**D-F**) Group size (**D**), mean resource value of individuals on peaks (**E**), and mean resource value of all individuals (**F**) as a function of the mean environmental sensitivity parameter $\psi_1$ of a population perturbed from ESSt. Orange point in (**F**) indicates values corresponding to ESSt. Note rapid decrease in mean fitness for perturbations in both directions. Semitransparent points are results of 2000 individual simulation runs. To compute means and standard errors, simulation runs were divided into 50 evenly spaced bins. Bolded points and error bars show mean of each bin $\pm$ 2 standard errors.

## 2.8 Evolution with resource consumption

As described in the Main Text, social interactions confer a fitness advantage to social individuals at least in part because the positions and local densities of a given individual's neighbors contain information about the spatial distribution of resources. However, if individuals quickly consume resources, this may break down. For example, areas in which the density of neighbors is currently high may no longer contain resources in the near future if those neighbors consume the resources. To explore how resource consumption affects evolutionary dynamics, we repeated evolutionary simulations assuming individuals consume the resource. To model resource consumption, we assume each individual consumes resources at a rate given by the product of the resource value at its position $S(\mathbf{x}_i)$ and a consumption rate constant $u$. At each time step, the height of peak $j$, $\lambda_{0,j}$, is reduced by the sum $\sum_{i=1}^{N} uS(\mathbf{x}_i, \mathbf{x}_s)$, where $\mathbf{x}_s$ is the location of the peak and $N$ is the number of individuals in the population. We

assume individuals abandon a resource when $\lambda_{0,j}$ falls below $s^*$. To keep the number of resource peaks constant and the total amount of resource on the landscape from being completely depleted, we allow resource peaks that reach a height of $\lambda_{0,j} = s^*$ to regenerate at a new location chosen at random with equal probability from all points in the environment. The new peak has a peak height equal to the starting peak height, $\lambda_0$. Mean resource value for each agent is calculated in the same manner as in the case where peaks are not depleted.

*Appendix Figure 7* shows the results of replicate evolutionary simulations with high (A) and low (B) rates of resource consumption. In the case of high consumption (*Appendix Figure 7A*), individuals evolve to have increasing mean values of $\psi_0$ and $\psi_1$, and $\psi_0$ values are well above the hysteresis regime between collective states. While values of $l_{max}$ still enable individuals to be attracted to one another at intermediate to large distances, the variation in $l_{max}$ values among replicate simulations suggests that there is not strong selection for large $l_{max}$. When individuals consume resources at a lower rate (*Appendix Figure 7B*), results parallel those shown in *Figure 1* of the Main Text; populations evolve mean values of $\psi_0$ that are directly above the hysteresis region, $\psi_1$ approaches a stable value, and $l_{max}$ approaches the maximum allowable value of 30.

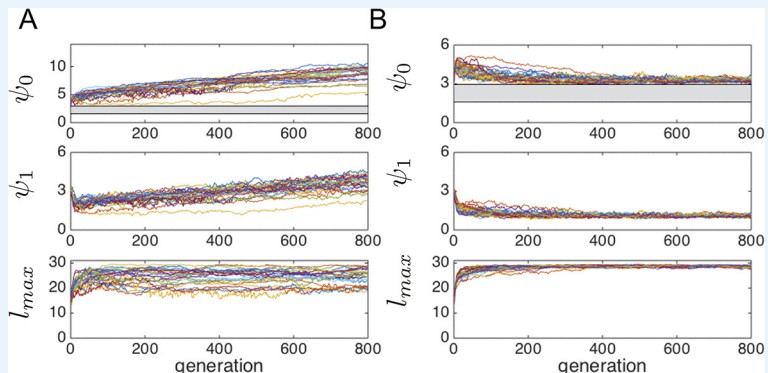

**Appendix Figure 7.** Evolution of behavioral traits when individuals consume resource. Lines show means of independent evolutionary simulations. (A) High consumption rate corresponding to fast depletion of resource peaks. (B) Intermediate consumption rate corresponding to slower depletion of the peaks. Note different axis limits in the top panels of **A** and **B**. Grey region corresponds to hysteresis region between collective states shown in Main Text **Figure 3**. Parameters are as follows: $s^*=2$, high consumption rate $= 3.2*10^{-3}$ (time step$^{-1}$), low consumption rate $= 8.0*10^{-5}$ (time step$^{-1}$), N = 300. All other parameters as in **Figure 1** of the Main Text. These consumption rates correspond to the case where 100 individuals near the peak center can deplete a peak in roughly five time steps (fast depletion, **A**), and the case where the same task takes 200 time steps (slower depletion, **B**).

# 3 The cohesive state is characterized by a fixed, finite density

Agents obeying the equations described in the Main text exhibit several distinct collective states. One such state, which we call the cohesive state, is characterized by dense groups of agents occupying a fixed fraction of the environment. One of the salient properties of these groups is that they eventually reach a fixed density that becomes independent of group size. Using a small number of simple assumptions about the behavior of agents within a cohesive group, we are able to predict the density of agents directly from the model parameters. The motivation of our calculation comes from the structure of the equations, which include social potential terms and velocity-dependent self-propulsion terms. The social force on an agent is

given by **Equation A1**. We will rewrite the social potential in **Equation A1** (i.e., the term in brackets) as

$$\Phi(\mathbf{x}_i) = \sum_{j \in N_i} [-C_a e^{-|\mathbf{x}_i - \mathbf{x}_j|/l_a} + C_r e^{-|\mathbf{x}_i - \mathbf{x}_j|/l_r}] \tag{A4}$$

The effect of the potential term in the equations is to exert a force on the entire system towards configurations where the potential energy is lower. The propulsive forces are non-conservative, causing phase-space volumes to contract, allowing the system to approach a potential energy minimum. We model the cohesive state as $N$ agents occupying a circular region of radius $l$. Further, we assume that the probability distribution of agents within this circular region is uniform, so that agent density is given by:

$$\rho = \frac{N}{\pi l^2} \tag{A5}$$

The density $\rho$ lets us define an interaction radius $l_I$ which is expected to contain $k$ individuals:

$$l_I = l\sqrt{\frac{k}{N}} \tag{A6}$$

This expression is valid when $N > k$, which is the case we are interested in. When $N < k$ the interaction radius is simply the group radius, $l = l_I$, and each agent interacts with every other agent. We calculate the expected potential by integrating over a circle of radius $l_I$:

$$\Phi_i = \rho \int_{|\mathbf{x}_i - \mathbf{x}| < l_I} \left( -C_a e^{-|\mathbf{x}_i - \mathbf{x}|/l_a} + C_r e^{-|\mathbf{x}_i - \mathbf{x}|/l_r} \right) dA \tag{A7}$$

This integral evaluates to the following expression:

$$\Phi_i = \frac{-2C_a N}{l^2}\left[ l_a^2 - \left( l_a^2 + \sqrt{\frac{k}{N}}l l_a \right)e^{-\sqrt{\frac{k}{N}}\frac{l}{l_a}} \right] + \frac{2C_r N}{l^2}\left[ l_r^2 - \left( l_r^2 + \sqrt{\frac{k}{N}}l l_r \right)e^{-\sqrt{\frac{k}{N}}\frac{l}{l_r}} \right] \tag{A8}$$

It is illustrative to write this expression after the substitution $l = \sqrt{\frac{N}{\pi\rho}}$:

$$\Phi_i = -2\pi\rho \left[ C_a \left( l_a^2 - e^{-\sqrt{\frac{k}{\pi\rho l_a^2}}}\left[ l_a^2 + l_a\sqrt{\frac{k}{\pi\rho}} \right] \right) - C_r \left( l_r^2 - e^{-\sqrt{\frac{k}{\pi\rho l_r^2}}}\left[ l_r^2 + l_r\sqrt{\frac{k}{\pi\rho}} \right] \right) \right] \tag{A9}$$

The density that minimizes $\Phi_i$ will be the density of agents in the cohesive state. When written this way, it is clear that $N$ will not influence the location of the minimum of $\Phi_i$, as $N$ only appears in the expression as a constant multiplier. Thus, when $N > k$ we expect cohesive groups to have a constant density, so that the radius of a group grows like $\sqrt{N}$. These predictions match the results of our simulation quite closely, which one can see from comparisons between **Appendix Figure 8** to the lower branch of the hysteresis plot in **Appendix Figure 9**.

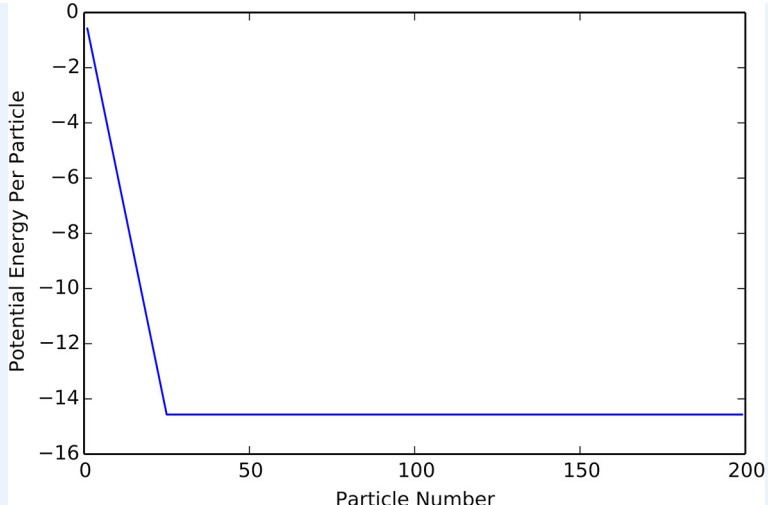

**Appendix Figure 8.** Plot of potential energy per agent as a function of group size. Cutoff at *N* = *K* indicates constant density.

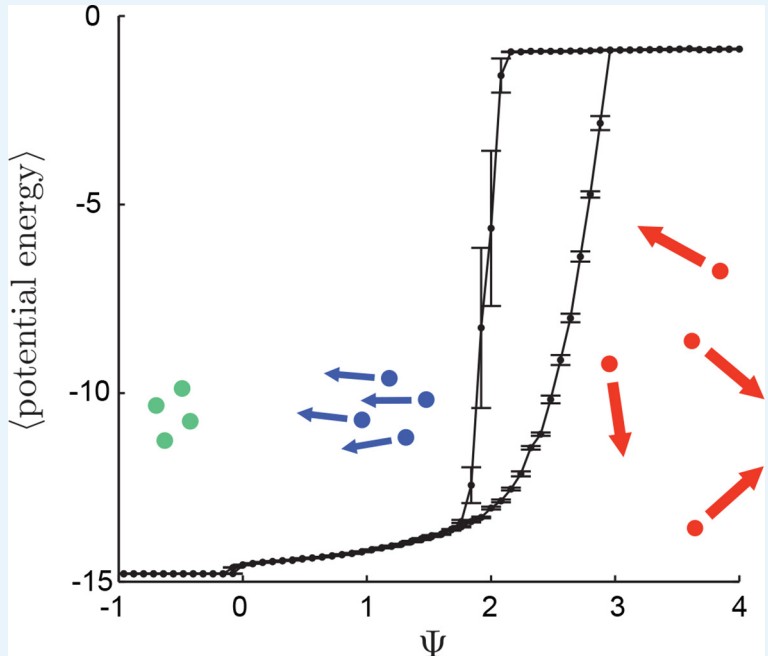

**Appendix Figure 9.** Hysteresis plot of potential energy averaged over the entire population (compare with *Figure 3* in Main Text).
Figure produced by starting with a population with $\Psi$ = 4 in a uniform environment. Population is allowed to equilibrate for 5,000 time steps and then average potential energy is calculated using *Equation 4* in the Main Text. $\Psi$ is then lowered. This process is repeated until $\Psi$ = −1, at which point the same procedure is used to increase $\Psi$. Upper curve corresponds to decreasing $\Psi$. Lower curve corresponds to increasing $\Psi$. Note drop in mean potential energy at $\Psi$ = 0. We refer to the states on either side of this transition as *station-keeping* ($\Psi$ < 0) and *cohesive* $\Psi$ > 0 *and below upper transitional regime at* $\Psi$ ~ 1.7. Points and (error bars) correspond to mean (and 2 standard errors) of 50 replicate simulations. Parameters as in *Figure 1* of the Main Text with $l_{max}$ = 30.

That density is necessarily constant with increasing $N$ is a hallmark of topological interaction laws which are repulsive at short range. When there is no restriction on the number of interaction neighbors, an interaction law of the type that we use can give rise to catastrophic behavior, where the group density increases without bound with increasing $N$ (**D'Orsogna et al., 2006**). One feature of the topological interaction is that it allows for biological realism for agent parameter values that would otherwise lead to catastrophic behavior.

## 4 Relationship between $\psi_0$ and the relative strength of social forces

In order to better understand how changing parameters affect our model, we ignore stochasticity and consider the equations for the acceleration and velocity in a homogeneous environment without resources (so that the background velocity is constant). First we define:

$$v_0 = \sqrt{\frac{\psi_0}{\eta}} \tag{A10}$$

The equations become:

$$\frac{d\mathbf{x}_i}{dt} = \mathbf{v}_i \tag{A11}$$

$$\frac{d\mathbf{v}_i}{dt} = \eta\left(|v_0|^2 - |v_i^2|\right)\frac{\mathbf{v}_i}{|v_i|} + \nabla\Phi_i \tag{A12}$$

Let $l_0$ be a characteristic length scale in this problem, let $v_0$ be a characteristic velocity scale, and let $t_0 = \frac{l_0}{v_0}$ be a characteristic time scale. Then, let $C_a$, the attraction coefficient, be the scale of the potential. We non-dimensionalize our equations by rewriting them in terms of the dimensionless variables:

$$x' = \frac{x}{l_0} \quad t' = \frac{t}{t_0} \quad v' = \frac{v}{v_0} \quad \Phi' = \frac{\Phi}{C_a} \tag{A13}$$

The resulting dimensionless equations are:

$$\frac{dx'}{dt'} = v' \tag{A14}$$

$$\frac{dv'}{dt'} = t_0|v_0|\eta\left(1 - |v'|^2\right)\frac{v'}{|v'|} + \frac{l_0^2 C_a}{v_0^2}\nabla'\Phi_i' \tag{A15}$$

The non-dimensional number $\frac{l_0^2 C_a}{v_0^2} = \frac{\eta l_0^2 C_a}{\psi_0}$ measures the relative strength of the social potential. When $\psi_0$ becomes large, social forces become negligible. The reason for this effect is that agents begin moving too quickly for the social forces to have any appreciable effect on their trajectories. Therefore, for constant $l_0$ and $C_a$, an alternative interpretation of $\psi_0$ is as a term that dictates the relative strengths of autonomous versus social forces.

# 5 Continuum description predicts cohesive and dispersed state

*Figure 3* in the Main Text illustrates that populations exhibit distinct regimes, which we refer to as collective states, as a function of the preferred speed parameter. Two states are evident in *Figure 3* in the Main Text: a state with short inter-individual distances for small $\Psi$ and a state with large inter-individual distances for large $\Psi$. A third state is evident if the mean potential energy of all individuals in the population is plotted as a function of $\Psi$ in a uniform environment, where potential energy is calculated from *Equations 3 and 4* in the Main Text. For $\Psi < 0$ there is a distinct drop in potential energy for decreasing $\Psi$ (*Appendix Figure 9*). We refer to the state that occurs for $\Psi < 0$ as *station-keeping* in the Main Text.

In order to better understand the behavior of agents in the context of our model, we have developed a continuum equation for the time evolution of agent density. In the context of a homogeneous environment this description can be used to predict the points in parameter space at which the uniform, purely solitary state becomes unstable, and to demonstrate that heterogeneous states cannot be stable at high enough background velocity. Although the continuum description is only an approximation, it is able to qualitatively predict many of the features of our multi-agent simulations, which makes the mechanisms responsible for this behavior mathematically more explicit. In order to derive continuum equations, we begin with a Liouville equation for the probability density of all the $N$ particles within the full phase space, and derive a hierarchy of equations by taking moments (*Flierl et al., 1999*; *Born and Green, 1946*). This hierarchy can be closed by assuming that stochastic forces are sufficiently strong to ensure independence of the individual agents. For the analysis presented here, we assume that the agents travel at a constant velocity $v_0 = \sqrt{\Psi}$ (using the angular variable $\theta$ to describe the direction of the velocity), and that there is noise in the angular velocity driven by a Wiener process with variance $\sqrt{2\varepsilon}$ per unit time. The assumption of a constant velocity implies that we have taken a limit where $\eta$, $\psi_0$, and $\psi_1$ go to infinity, though their ratios remain constant. We will let $\psi_0$ and $\psi_1$ stand for the limiting ratios of the original model. We will denote the space of positions by $V$, which will be a 2-torus with length $L_D$. The assumption of a constant velocity and of angular noise lead only to small quantitative changes in agent behavior, and they make it possible to analyze the resulting equations.

Therefore we begin with the following set of stochastic differential equations:

$$dx_i = v_0 \cos(\theta_i) dt \tag{A16}$$

$$dy_i = v_0 \sin(\theta_i) dt \tag{A17}$$

$$d\theta_i = \frac{|F_i(\mathbf{x})|}{v_0} \sin(\Gamma_i(\mathbf{x}) - \theta_i) dt + \sqrt{2\varepsilon} dW \tag{A18}$$

Here $F_i$ is the social force on agent $i$, and $\Gamma_i$ is the angular direction of the social force on agent $i$. We assume that this force is produced through a topological interaction of the following form:

$$F_i = \sum_{j \in N_i} -\nabla w(|x_i - x_j|) \tag{A19}$$

Here $N_i$ is the set of $k$ closest neighbors to agent $x_i$, and $w$ is an interaction kernel.

From these equations, we are able to write a Liouville equation by introducing a probability density on phase space, $P(\{\mathbf{x}\}, \{\theta\})$, where $\{\mathbf{x}\}$ is the set of $N$ agent positions and $\{\theta\}$ is the set of $N$ agent directions. The value of $P$ at a given set of positions and directions is the

probability that the each of the agents have the specified positions and directions. The Liouville equation is:

$$\frac{\partial P}{\partial t} + v_0 \sum_{i=1}^{N} \hat{e}_{\theta_i} \cdot \nabla_i P + \frac{1}{v_0} \sum_{i=1}^{N} \frac{\partial}{\partial \theta_i}$$
$$\left( P \sum_{j \in N_i(\{\theta\})} |F(|\mathbf{x}_i - \mathbf{x}_j|)| \sin\left(\Gamma(\mathbf{x}_i - \mathbf{x}_j) - \theta_i\right) \right) \tag{A20}$$
$$= \varepsilon \sum_{i=1}^{N} \frac{\partial^2 P}{\partial \theta_i^2}$$

In order to simplify this equation, we assume that the probability density function $P$ can be factorized into the product of $N$ identical single particle probability density functions:

$$P(\{\mathbf{x}\},\{\theta\}) = \prod_{i=1}^{N} p(\mathbf{x}_i,\theta_i) \tag{A21}$$

This assumption is equivalent to assuming statistical independence of the positions and directions of the agents, a condition which could be reached either through large stochasticity $\varepsilon$ or ergodic single particle trajectories. The assumption allows us to derive a closed equation for the single particle probability density function $p$, in a similar fashion to a closure of the usual BBGKY hierarchy in kinetic theory.

Then, we are able to write the following equation for the single agent probability density function $p(x,\theta,t)$ (where we have replaced a binomial distribution with a Poisson distribution):

$$\frac{\partial p}{\partial t} + v_0 \hat{e}_\theta \cdot \nabla_x p + \frac{1}{v_0} \frac{\partial}{\partial \theta} \left( \int_{V \times S^1} d^2x' d\theta' G(\mathbf{x},\mathbf{x}',\theta) p(x,\theta) \right)$$
$$= \varepsilon \frac{\partial^2 p}{\partial \theta^2} \tag{A22}$$

Here, the expression $\lambda(\mathbf{x},\mathbf{x}')$ is the expected number of agents within a distance $|\mathbf{x}'|$ from the point $\mathbf{x}$, the function $w(|\mathbf{x}|)$ is the social potential between two particles, and the expression $\Gamma(\mathbf{x},\mathbf{x}')$ is the angle of force from an agent located at $\mathbf{x}'$ to an agent located at $\mathbf{x}$:

$$G(\mathbf{x},\mathbf{x}') = \sum_{j=0}^{k-1} \frac{e^{-\lambda(\mathbf{x},\mathbf{x}')} \lambda(\mathbf{x},\mathbf{x}')^j}{j!} |\nabla w(|\mathbf{x} - \mathbf{x}'|)| \sin(\Gamma(\mathbf{x},\mathbf{x}') - \theta) \tag{A23}$$

$$\lambda(\mathbf{x},\ \mathbf{x}') = \int_{|\mathbf{x}''| < |\mathbf{x} - \mathbf{x}'|} \int_0^{2\pi} d\theta d^2x'' N p_1(\mathbf{x} + \mathbf{x}'',\theta) \tag{A24}$$

$$w(x) = -C_a e^{-|\mathbf{x}|/l_a} + C_r e^{-|\mathbf{x}|/l_r} \tag{A25}$$

$$\Gamma(\mathbf{x},\ \mathbf{x}') = \arg(\nabla_x w(|\mathbf{x} - \mathbf{x}'|)) \tag{A26}$$

The presence of the terms involving $\lambda$ are a consequence of the topological interaction law between the agents. This equation is most accurate in the limit where $\rho_c l_c^2 \gg 1$, where $\rho_c$ is a characteristic density and $l_c$ is the characteristic length scale of the interaction. In the examples which we considered in our study, this ratio is typically only slightly larger than 1 (see *Peshkov et al., 2012*; *Chou et al., 2012*) for derivations of continuum descriptions of topological interactions in a more collisional regime). Despite that, we find both quantitative and qualitative agreement between the continuum description and the agent based model.

This kinetic description can be converted into a hierarchy of fluid equations by taking moments with respect to the angular direction variable $\theta$.

We introduce the phase space particle density $f$:

$$f = Np \tag{A27}$$

The Fourier series for $f$ gives the important macroscopic variables, for instance:

$$\rho = \int_0^{2\pi} f d\theta \quad \rho u_x = \int_0^{2\pi} f v_0 cos(\theta) d\theta \quad \rho u_y = \int_0^{2\pi} f v_0 sin(\theta) d\theta \tag{A28}$$

Here $\mathbf{u}$ is the mean velocity of agents at each point in space. We will take moments of the kinetic equation through Fourier series:

$$f(\mathbf{x}, \theta) = \sum_{m=-\infty}^{\infty} \tilde{f}_m(\mathbf{x}) e^{im\theta} \tag{A29}$$

The time evolution equation for the $m$th Fourier coefficient is:

$$\begin{aligned}
\frac{\partial \tilde{f}_m}{\partial t} + \frac{v_0}{2}\left[\left(\frac{\partial}{\partial x} + i\frac{\partial}{\partial y}\right)\tilde{f}_{m+1} + \left(\frac{\partial}{\partial x} - i\frac{\partial}{\partial y}\right)\tilde{f}_{m-1}\right] \\
+ \frac{m|\mathbf{F}(\mathbf{x},\,\rho)|}{2v_0}\left(e^{i\Theta(\mathbf{x},\,\rho)}\tilde{f}_{m+1} - e^{-i\Theta(\mathbf{x},\,\rho)}\tilde{f}_{m-1}\right) = -m^2\varepsilon\tilde{f}_m
\end{aligned} \tag{A30}$$

Here, we have simplified this expression by introducing two new functionals of the density, $F(\mathbf{x},\rho)$ and $\Theta(\mathbf{x},\rho)$, which represent the social force exerted at the point $\mathbf{x}$ due to the density $\rho$ and the direction of that social force. Explicitly these are given by:

$$\mathbf{F}(\mathbf{x},\rho) = -\int_V d^2 x' \sum_{j=0}^{k-1} \frac{e^{-\lambda(\mathbf{x},\mathbf{x}')}\lambda(\mathbf{x},\mathbf{x}')^j}{j!} \nabla w(|\mathbf{x}-\mathbf{x}'|)\frac{\rho(\mathbf{x}')}{N} \tag{A31}$$

$$\Theta(\mathbf{x},\rho) = \arg(-\mathbf{F}(\mathbf{x},\rho)) \tag{A32}$$

The evolution of the $m$th moment depends on the value of the $m+1$st moment, so that we have an infinite hierarchy of equations. Moments with high values of $|m|$ experience strong damping, and we can use this to justify discarding all moments with $|m|$ above a given threshold. In the following treatment we will set to zero all Fourier coefficients with $|m| > 1$, which is the simplest truncation of the hierarchy that leads to non-trivial equations.

$$\frac{\partial \rho}{\partial t} + \nabla\cdot(\rho\mathbf{u}) = 0 \tag{A33}$$

$$\frac{\partial \rho\mathbf{u}}{\partial t} + \frac{v_0^2}{2}\nabla\rho - \frac{1}{2}\mathbf{F}(\mathbf{x},\rho)\rho = -\varepsilon\rho\mathbf{u} \tag{A34}$$

The right hand side of the momentum equation leads to rapid equilibration, and we can eliminate the time derivative in this equation. This allows us to find an expression for $\rho\mathbf{u}$ in terms of $\rho$ only:

$$\rho\mathbf{u} = -\frac{v_0^2}{2\varepsilon}\nabla\rho + \frac{1}{2\varepsilon}\mathbf{F}(\mathbf{x},\rho)\rho \tag{A35}$$

We can use this to write a single closed equation for $\rho$. In order to facilitate the analysis of this equation, we make one further approximation, replacing the sum over Poisson factors with a Heaviside function that is equal to 1 if the expected number of agents between inside a ball of size $x'$ is less than $k$, and 0 otherwise. This captures the dominant qualitative feature of the topological interaction in a simple way: the effective interaction radius is a function of the local density. This approximation is quantitatively consistent with the assumptions of the previous section and the results of our simulations. The resulting equation is:

$$\frac{\partial \rho}{\partial t} + \nabla \cdot \left( -\frac{v_0^2}{2\varepsilon} \nabla \rho + \frac{\rho}{2\varepsilon} \int_{|\mathbf{x}-\mathbf{x}'|<L(\rho,\mathbf{x})} \nabla w(|\mathbf{x}-\mathbf{x}'|)\rho(\mathbf{x}')d^2x' \right) = 0 \qquad \text{(A36)}$$

$$L(\rho,\mathbf{x}) = \sqrt{\frac{k}{\pi\rho(\mathbf{x})}} \qquad \text{(A37)}$$

The advection-diffusion equation will be used to understand the behavior of our multi-agent simulations. From its form one can see the effects of $v_0$: large $v_0$ enhances the diffusivity and reduces the effects of the potential.

## 5.1 Formation of the cohesive state and stability of the dispersed state

Any constant density function $\rho_0$ is an equilibrium solution of **Equation A36**. A crucial property of our multi-agent model is that depending on the background velocity, the agents have the ability to spontaneously form a dense state which we call the cohesive state. In order for this to be possible, the uniform state must be unstable. One advantage of the continuum description is that it allows us to investigate such questions within a much simpler framework than in the original agent based model. In order to do so, we select a uniform state with value $\rho_0$ and linearize around it, neglecting terms second order or higher in the deviation away from $\rho_0$:

$$\frac{\partial \rho_1}{\partial t} = \nabla \cdot \left( \frac{v_0^2}{2\varepsilon} \nabla \rho_1 - \frac{\rho_0}{2\varepsilon} \int_{|\mathbf{x}-\mathbf{x}'|<L_0} \nabla w(|\mathbf{x}-\mathbf{x}'|)\rho_1(\mathbf{x}')d^2x' \right) \qquad \text{(A38)}$$

$$L_0 = \sqrt{\frac{k}{\pi\rho_0}} \qquad \text{(A39)}$$

This is translationally invariant, and we have periodic boundary conditions, so we consider the Fourier coefficients of $\rho_1$:

$$\frac{\partial \tilde{\rho}_{1,\mathbf{j}}}{\partial t} = -\tilde{\rho}_{1,\mathbf{j}} \left( |\mathbf{j}|^2 \frac{2\pi^2 v_0^2}{\varepsilon L_D^2} - G(|\mathbf{j}|) \right) \qquad \text{(A40)}$$

The term $G(|\mathbf{j}|)$ can be calculated by application of the convolution theorem and integration by parts, using the fact that the integrals are radially symmetric (which is true as long as $L_0 < L_D$):

$$G(|\mathbf{j}|) = \frac{-\rho_0 \pi i \mathbf{j}}{\varepsilon L_D} \cdot \int_V e^{\frac{-2\pi i \mathbf{j} \cdot x}{L_D}} \nabla w(|\mathbf{x}|) H(L_0 - |\mathbf{x}|) d^2x \qquad \text{(A41)}$$

$$= -\int_0^{L_0} \frac{2\pi^2 \rho_0 |\mathbf{j}|}{\varepsilon L_D} dr r \frac{\partial w}{\partial r} J_1\left( \frac{2\pi|\mathbf{j}|r}{L_D} \right) \qquad \text{(A42)}$$

Here $J_1$ stand for the corresponding Bessel functions of the first kind. Linear stability is determined by the sign of the coefficient on $\rho_{\tilde{1},j}$ on the right hand side of the following expression:

$$\frac{\partial \rho_{\tilde{1},j}}{\partial t} = -\frac{2\pi^2 v_0^2 |\mathbf{j}|^2}{\varepsilon L_D^2} \rho_{\tilde{1},j} \left(1 - \frac{L_D \rho_0}{|\mathbf{j}| v_0^2} \int_0^{L_0} r\, dr \frac{dw(r)}{dr} J_1\left(\frac{2\pi |\mathbf{j}| r}{L_D}\right)\right) \tag{A43}$$

We can use our formula to determine the stability or instability of an arbitrary homogeneous equilibrium solution. We have plotted an example of this in **Appendix Figure 10**. A number of general features emerge from these diagrams. Increases in the background velocity $v_0$ always promotes stability of the dispersed state. The agents make use of this feature to enable themselves to transition from the dispersed state to the cohesive state in regions where $\Psi$ crosses below the stability threshold. Increases in the number of interaction neighbors $k$, the decay length of the attractive interaction $l_a$, and the strength of the attractive interaction $C_a$ all promote instability of the dispersed state, though at large $k$ further increases in $k$ have little effect. The background density of agents has a more complicated effect on the stability of the dispersed state, when $\rho_0$ is low the social forces are very weak because the distance between agents is large, so that a very small $v_0$ is required for formation of the dispersed state. When $\rho_0$ gets too large, the repulsive part of the interaction becomes more important and stability of the dispersed state is promoted.

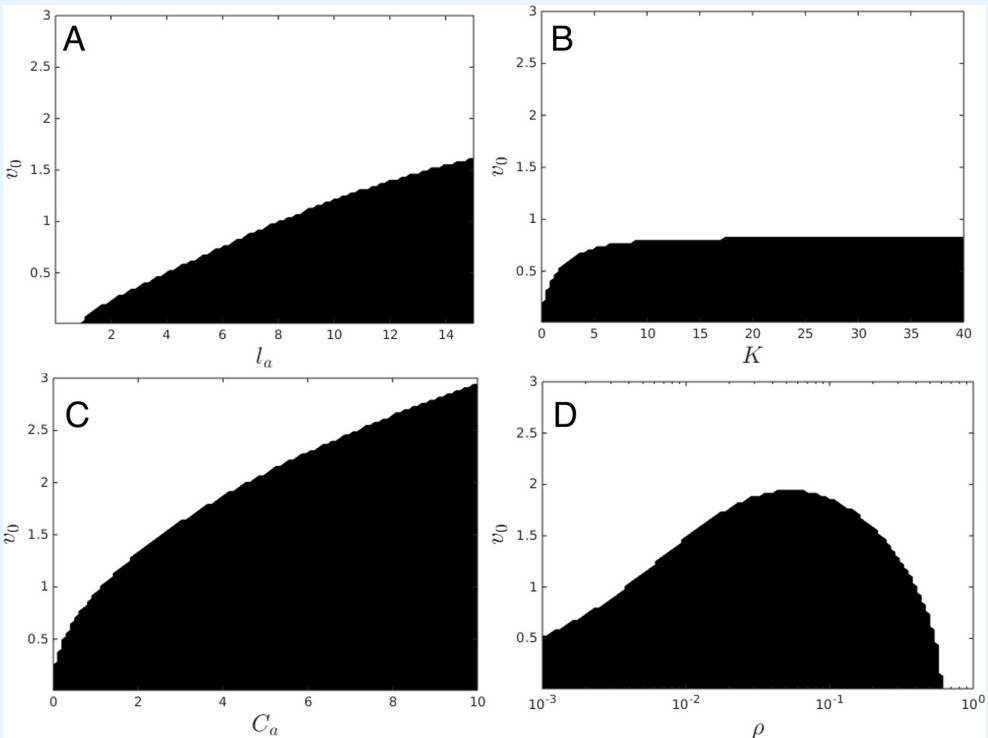

**Appendix figure 10.** Regions of parameter space where *dispersed* state is stable (white) and unstable (black).

Instability of dispersed state causes transition to *cohesive* state. (**A**) Sensitivity to $l_a$, for $\rho$ = .0025, $K$ = 25, $l_r$ = 1.0, $C_a$ = 1.0, $C_r$ = 1.1. Increasing $l_a$ promotes instability and formation of the *cohesive* state, as does decreased $v_0$. (**B**) Sensitivity to $K$, for $\rho$ = .0025, $l_a$ = 7.5, $l_r$ = 1.0, $C_a$ = 1.0, $C_r$ = 1.1. Increasing $K$ promotes instability and formation of the *cohesive* state, as does decreased $v_0$. Large values of $K$ lead to roughly the same stability properties, due to the exponential decay of the interaction with length. (**C**) Sensitivity to $C_a$, for $\rho$ = .0025, $l_a$ = 7.5, $l_r$ = 1.0, $K$ = 25, $C_r$ = 1.1. Increasing $C_a$ promotes instability and formation of the cohesive state,

as does decreased $v_0$. (**D**) Sensitivity to $\rho$. Smaller $v_0$ promotes instability. When $\rho$ is very small, increasing $\rho$ makes instability more likely.

## 5.2 Nonlinear stability of the dispersed state for high $v_0$

*Equation A36* combines a diffusive term with effective diffusion coefficient $\frac{v_0^2}{2\varepsilon}$, and a term due to the social forces, which is proportional to the magnitude of the social forces and $\frac{1}{2\varepsilon}$. On the basis of this, we expect that as $v_0$ increases the diffusive terms become more important relative to the social forces. In the linear stability analysis, this manifested itself through instability of the homogeneous base state when $v_0$ was decreased below a threshold. When $v_0$ is large enough, we are able to prove using energy inequalities (*Doering and Gibbon, 1995*) that the homogeneous equilibrium state is a global attractor. The implication of this is that above a certain threshold the cohesive state can no longer exist, and all the agents enter the dispersed state. Combined with the results of the previous subsection, this provides an analytical demonstration of the hysteresis that we observe in our multi-agent model.

In order to establish these results, we rewrite the dynamical equation for $\rho$ in terms of the deviation from the mean density $\overline{\rho} = \int_V \frac{\rho}{L_D^2} d^2x$. We define $\rho_1 = \rho - \overline{\rho}$. Then the equation for $\rho_1$ is:

$$\frac{\partial \rho_1}{\partial t} + \nabla \cdot \left( -\frac{v_0^2}{2\varepsilon}\nabla\rho_1 + \frac{(\rho_1 + \overline{\rho})}{2\varepsilon}\int_{|\mathbf{x}-\mathbf{x}'|<L(\mathbf{x})} \nabla w(|\mathbf{x}-\mathbf{x}'|)\rho_1(\mathbf{x}')d^2x' \right) = 0 \tag{A44}$$

We multiply by $\rho_1$ on both sides of the equation and integrate:

$$\begin{aligned}
\frac{d\|\rho_1\|_2^2}{dt} &= -\frac{v_0^2}{2\varepsilon}\|\nabla\rho_1\|_2^2 - \int_V \frac{\rho_1\overline{\rho}}{2\varepsilon}\nabla\cdot\int_{|\mathbf{x}-\mathbf{x}'|<L} d^2x'\nabla w(|\mathbf{x}-\mathbf{x}'|)\rho_1(\mathbf{x}')d^2x \\
&\quad - \int_V \frac{\rho_1}{2\varepsilon}\nabla\cdot\left(\rho_1\int_{|\mathbf{x}-\mathbf{x}'|<L} d^2x'\nabla w(|\mathbf{x}-\mathbf{x}'|)\rho_1(\mathbf{x}')\right)d^2x
\end{aligned} \tag{A45}$$

Here the expression $\|f\|_p = \left(\int_V d^2x|f(\mathbf{x})|^p\right)^{1/p}$ is the standard $L^p$ norm on the space $V$. The first term on the right hand side can be bounded through use of the Poincare inequality for a mean zero function on the torus:

$$\|\nabla\rho_1\|_2^2 \geq \frac{4\pi^2}{L_D^2}\|\rho_1\|_2^2 \tag{A46}$$

The second term on the right hand side can be simplified by performing integration by parts in order to transfer the gradient operators onto the $\rho_1$ terms:

$$\left| \frac{1}{2\varepsilon}\overline{\rho}\int_V d^2x\rho_1(\mathbf{x})\nabla\cdot\int_{|\mathbf{x}-\mathbf{x}'|<L} d^2x'\nabla\omega(|\mathbf{x}-\mathbf{x}'|)\rho_1(\mathbf{x}') \right| \leq \frac{|\overline{\rho}|}{2\varepsilon}\|\nabla\rho_1\|_2\|\rho_1\|_2\|H(L-|\mathbf{x}|)\nabla\omega(|\mathbf{x}|)\|_1 \tag{A47}$$

Here we have made use of Young's Inequality, which states that $\|f*g\|_r \leq \|f\|_p \cdot \|g\|_q$ when $\frac{1}{r} = \frac{1}{p} + \frac{1}{q} - 1$.

The third term on the right hand side is the most difficult to deal with because it contains three powers of $\rho_1$. To bound this term we make use of the fact that the density $\rho$ is always positive, which implies that $\rho_1 > -\overline{\rho}$. Then, because $\rho_1$ has zero mean, we can bound $\|\rho_1\|_1$:

$$\|\rho_1\|_1 \leq 2\|\overline{\rho}\|_1 \tag{A48}$$

Using this expression (and Young's Inequality), we find that:

$$\left| \int_V d^2x \rho_1 \nabla \cdot \left( \rho_1 \int_{|\mathbf{x}-\mathbf{x}'|<L} d^2x' \nabla \omega(|\mathbf{x}-\mathbf{x}'|) \rho_1(\mathbf{x}') \right) \right| \leq 2 \left( \int_V d^2x \right) \overline{\rho} ||\nabla \omega||_\infty ||\nabla \rho_1||_2 ||\rho_1||_2 \qquad \text{(A49)}$$

Using these bounds, we can write a differential inequality for $\frac{d||\rho_1||_2^2}{dt}$:

$$\frac{d||\rho_1||_2^2}{dt} \leq \left( -\frac{\nu_0^2}{2\varepsilon} + \frac{L_D \overline{\rho} ||H(L-|\mathbf{x}|)\nabla w(|\mathbf{x}|)||_1}{4\pi\varepsilon} + \frac{L_D^3}{2\pi\varepsilon}\overline{\rho}||\nabla w||_\infty \right) ||\nabla \rho_1||_2^2 \qquad \text{(A50)}$$

If $v_0$ satisfies the following inequality:

$$\nu_0 \geq \sqrt{\frac{LD}{4\pi}\overline{\rho} \left( ||H(L-|\mathbf{x}|)\nabla w(|\mathbf{x}|)||_1 + 2L_D^2||\nabla w||_\infty \right)} \qquad \text{(A51)}$$

then the coefficient of $||\nabla \rho_1||_2^2$ in the differential inequality is negative, and we can apply the Poincare inequality to find:

$$\frac{d||\rho_1||_2^2}{dt} \leq \left( -\frac{2\pi^2 v_0^2}{\varepsilon L_D^2} + \frac{\pi \overline{\rho}||H(L-|\mathbf{x}|)\nabla \omega(|\mathbf{x}|)||_1}{L_D \varepsilon} + \frac{2\pi L_D}{\varepsilon}\rho||\nabla \omega||_\infty \right) ||\rho_1||_2^2 \qquad \text{(A52)}$$

This inequality allows us to use Gronwall's lemma (***Doering and Gibbon, 1995***) to prove $||\rho_1||_2$ that converges to 0 as a function of time, which implies that the homogeneous state is globally attractive for sufficiently large $v_0$.

## 5.3 Conclusion from continuum model

The simulations of the agent based model indicated that our agents possessed two properties: for small $v_0$ a cohesive state forms spontaneously, for large $v_0$ only dispersed states are possible, and for moderate values of $v_0$ both cohesive and dispersed states are possible. We were able to create a continuum model that demonstrates the mechanisms behind these numerical observations. We showed that for small $v_0$, homogeneous background states are linearly unstable to the formation of clumped states. For larger $v_0$, the homogeneous background states are linearly stable. Further, we showed that for sufficiently large $v_0$, the homogeneous state is globally attractive, so that clumped states are not possible.

## 5.4 Some additional properties of the transition between collective states: nucleation rates and hysteresis

In the theory of first order phase transitions, hysteresis often arises because there is a free energy penalty for small droplets of the stable phase. This leads to extremely low probabilities of critical droplet formation near the transition temperature. In order to test whether this effect is responsible for the hysteresis in our model equations, we performed long time numerical simulations using values of $\Psi$ within the hysteresis region, allowing us to estimate the nucleation rate of the cohesive phase. We performed replicate simulations with $10^3$ agents, restarting the simulations each time the agents were able to form the cohesive state. The results of this simulation are shown in ***Appendix Figures 11*** and ***12***, illustrating the super-exponential growth in the mean nucleation time as $\Psi$ increases. This growth in nucleation times corresponds to an increase in the minimum radius of a stable group. When $\Psi$ increases above 1.7, the expected time for nucleating the cohesive state becomes extremely large, leading to strong hysteresis. We also computed $\frac{1}{\sqrt{log\overline{T}}}$ to illustrate the approximate scaling of the nucleation time (***Appendix Figure 12***). Because we use a topological interaction, we do not necessarily expect this scaling to hold for much larger values of $\Psi$, as groups with $N < 25$ will have an increasing, rather than constant, potential energy per particle.

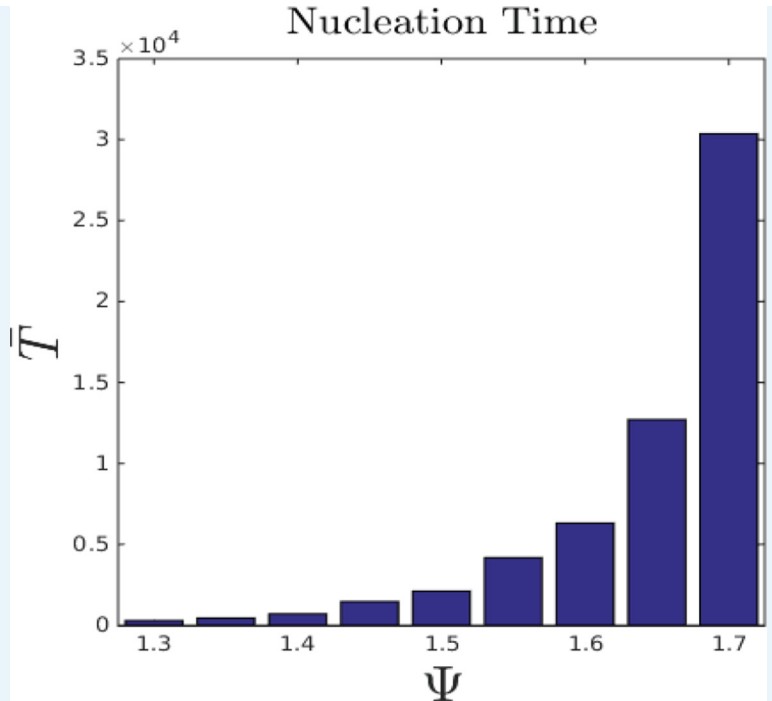

**Appendix Figure 11.** Mean nucleation time as a function of Ψ in the hysteresis region. Bars show mean nucleation time calculated from replicate simulations in which individuals begin the simulation with random starting positions. Simulation was ended when an agent reached a social potential value $< -14$, which is only possible if a dense group has formed. For each simulation, we denoted the time taken to satisfy this condition as the nucleation time. Parameter values taken from population at ESSt described above.

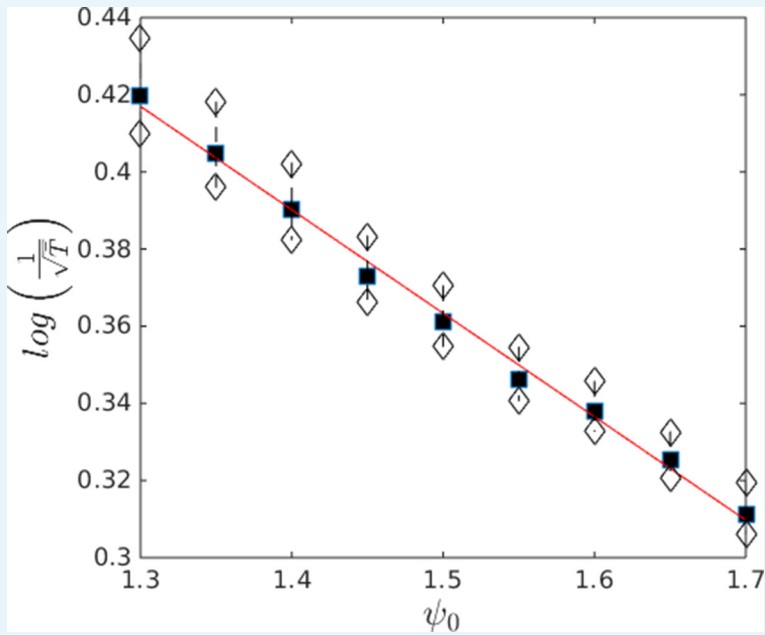

**Appendix Figure 12.** Approximate scaling of mean nucleation time with Ψ in the hysteresis region.
Data from **Appendix Figure 11**. Black points show means. White points are $\pm 1$ standard deviation. Note that the nucleation time is super-exponential in Ψ indicating that the

probability of nucleation becomes extremely small as $\Psi$ approaches the upper boundary of the hysteresis region.

# 6 Social gradient climbing and aggregation on a resource peak

In this section we derive a model of collective exploration and exploitation that allows us to understand how the ability of agents to find the resource peak changes when model parameters like $\psi_0$ or sociality are varied. We model agents as being either in a cohesive state near a resource peak or in a dispersed state. Using the model parameters and some simple assumptions about the dynamics, we calculate the fraction of particles approaching the resource peak that are able to enter the cohesive state. Using this model, we quantitatively estimate the rate at which agents are able to find the peak and the advantage of social agents versus asocial agents. We begin by stating a number of assumptions, each of which arises from some feature of our multi-agent model, that help make our theoretical analysis tractable.

## 6.1 Assumptions

1. The environmental response function is $\Psi(\mathbf{x}) = \psi_0 - \psi_1 A e^{-r^2/\sigma^2}$, indicating a single peak located at the origin and a preferred background velocity of $v_0 = \sqrt{\psi_0}$ in regions far from the resource peak.

2. Agents travel at the speed dictated by the environmental response function $\Psi$, so that $v(\mathbf{x}) = v(r) = \sqrt{\Psi}$. Only the direction of the velocity is allowed to vary.

3. Particles exist in one of two states, the cohesive state or the dispersed state. Particles in the cohesive state are close to the resource peak. Particles in the dispersed state have a uniform probability distribution in space and in direction. Particles in the cohesive state collectively produce a potential $\Phi_{\text{peak}}(r) = \min(k, \ N)C_a e^{-\frac{r}{l_a}}$.

4. Agents in the dispersed state interact only with particles in the cohesive state, and this interaction is cutoff for distances $r > l_{\max} = r_M$. The potential force is projected normal to the velocity of the agents so that it can only effect the agent directions.

5. An agent enters the cohesive state if it has a trajectory that reaches the radius of zero velocity, $r_t = \sigma \sqrt{\log\left(\frac{A\psi_1}{\psi_0}\right)}$, which is assumed to mark the transition between the cohesive and dispersed states.

## 6.2 Critical angle for capture by the peak

Consider a particle reaching $r_M$, the radius where it begins to feel the influence of the agents on the environmental resource peak, as is depicted in **Appendix Figure 13**. If the angle of the agent's trajectory is sufficiently directed towards the resource peak, the agent will reach the peak, and if the angle is directed sufficiently away, the agent will not reach the peak. There is a critical angle $\Delta_i$ at the boundary between these two scenarios. The size of $\Delta_i$ will determine the fraction of agents captured by the resource peak after crossing $r = r_M$, and it will also determine the flux of agents onto the resource peak.

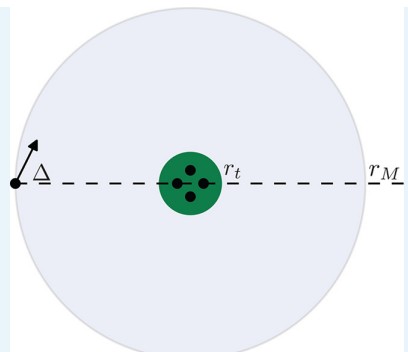

**Appendix Figure 13.** There is a group agents on the resource peak at the origin.
These agents are contained within the a circle of radius $r_t$, corresponding to the zero velocity region. An agent enters the region of radius $r_M$ and begins to feel the force from the agents on the peak. The angle of the velocity of the agent relative to the peak is $\Delta$.

To derive an expression for $\Delta_i$, we write equations for an agent traveling with a velocity of magnitude $v(r) = \sqrt{\psi_0 - \psi_1 A e^{-r^2/\sigma^2}}$, in direction $\omega$, experiencing the potential force $-\nabla\Phi_{\text{peak}}(r)$:

Our question is the following: *given initial radius $r = r_M$, initial angle $\theta_0$, and initial direction $\omega_0$, does the agent reach the zero velocity radius?* The equations of motion are:

$$\frac{dr}{dt} = v(r)\cos(\theta - \omega) \tag{A53}$$

$$\frac{d\theta}{dt} = \frac{v(r)}{r}\sin(\omega - \theta) \tag{A54}$$

$$\frac{d\omega}{dt} = \frac{\Phi'_{\text{peak}}(r)}{v(r)}\sin(\omega - \theta) \tag{A55}$$

We define the angle $\Delta = -\pi - \omega + \theta$, which is the angle between the velocity vector and the vector directed from the position of the agent to the origin. Then we can rewrite our equations in terms of the variables $r$ and $\Delta$ alone, leading to the following planar system:

$$\frac{dr}{dt} = -v(r)\cos(\Delta) \tag{A56}$$

$$\frac{d\Delta}{dt} = -\left(\frac{\Phi'_{\text{peak}}(r)}{v(r)} - \frac{v(r)}{r}\right)\sin(\Delta) \tag{A57}$$

The system of equations described here has the following properties:

1. If $\Delta = 0$, then $\frac{d\Delta}{dt} = 0$, so that $\Delta = 0$ for all further times. Similarly, $\Delta = 0$ implies $\frac{dr}{dt} < 0$, so that any agent with $\Delta = 0$ will reach the zero velocity radius.

2. If $-\frac{\pi}{2} < \Delta < \frac{\pi}{2}$, then $\frac{dr}{dt} < 0$ and the agent will move closer to the peak.

3. If both $\left(\frac{\Phi'_{peak}(r)}{v(r)} - \frac{v(r)}{r}\right) > 0$ and $-\frac{\pi}{2} < \Delta < \frac{\pi}{2}$, then $\frac{d|\Delta|}{dt} < 0$, and $\Delta$ becomes closer to $0$.

We make one additional assumption, which has been true in most practical cases, that allows us to make progress with the analysis.

**Assumption**: The function $\Phi'_{\text{peak}}(r) - \frac{v(r)^2}{r}$ has exactly one sign change on the interval $(r_0, \infty)$.

The location of the sign change occurs when $\Phi'_{\text{peak}}(r) = \frac{v(r)^2}{r}$, a point which we denote by $r^*$. At $r = r^*$, there is a balance between centrifugal and potential forces. We have the following inequalities:

$$\frac{\Phi'_{\text{peak}}(r)}{v(r)} < \frac{v(r)}{r} \text{ for } r < r*, \quad \frac{\Phi'_{\text{peak}}(r)}{v(r)} > \frac{v(r)}{r} \text{ for } r > r* \tag{A58}$$

This assumption allows us to divide the $(r, \Delta)$ plane into four regions:

1.
$$-\frac{\pi}{2} < \Delta < \frac{\pi}{2} \quad \text{and} \quad r < r*$$

2.
$$\Delta < -\frac{\pi}{2} \quad \text{or} \quad \frac{\pi}{2} < \Delta \quad \text{and} \quad r > r*$$

3.
$$-\frac{\pi}{2} < \Delta < \frac{\pi}{2} \quad \text{and} \quad r > r*$$

4.
$$\Delta < -\frac{\pi}{2} \quad \text{or} \quad \frac{\pi}{2} < \Delta \quad \text{and} \quad r < r*$$

In region 1, the agents move towards the peak and the potential force is stronger than the centrifugal force. In region 2, the agents move away from the peak, and the potential force is weaker than the centrifugal force. In region 3, agents move towards the peak but the potential force is weaker than the centrifugal force. In region 4, the agents move away from the peak but the potential is stronger than the centrifugal force. We conclude that:

1. Any trajectory that enters region 1 will reach the zero velocity radius.

2. Any trajectory that enters region 2 will escape to $\infty$.

Consider again the hypothetical agent in region 3 and at radius $r = r_M$. The agent will eventually either reach region 1, region 2, or the boundary points between the two regions (which are unstable equilibria).

1. The points $\left(r*, \pm\frac{\pi}{2}\right)$ are unstable equilibrium points, each corresponding to a periodic orbit around the resource peak.

2. There are two values of $\Delta$ such that the solution of the initial value problem with initial condition $(r_M, \Delta)$ reach these equilibria. We call these angles $\pm\Delta_i$, where we define $\Delta_i > 0$. Any trajectory with initial value $(r_M, \Delta)$, with $|\Delta| < \Delta_i$ will enter region I and be captured by the resource peak, and any trajectory with initial value satisfying $|\Delta| > \Delta_i$ will enter region II and escape the resource peak.

The angle $\Delta_i$ is the critical angle that we seek.

## 6.3 Solving the reduced system

Instead of considering the time dependent differential equation, we search for an equation that describes the shape of a trajectory, that is, we assume $\Delta$ is a single valued function of $r$, and use the original equation in combination with the chain rule to write a differential equation for $\Delta(r)$. This method is valid in regions where $\Delta$ is actually a single-valued function of $r$, and for

this to be the case integration must be restricted to regions where $\Phi'_{\text{peak}}(r) - \frac{v^2(r)}{r}$ has only one sign. The resulting equation will be valid only while a trajectory is in region $3$, and the coefficients of this equation will blow up at the border of region $3$.

The quotient of **Equation A56 and A57** is the desired equation:

$$\frac{dr}{d\Delta} = \frac{\cot(\Delta)}{\frac{\Phi'_{\text{peak}}(r)}{v(r)^2} - \frac{1}{r}}$$

(A59)

**Equation A59** can be solved by integrating along a trajectory beginning at $(r_M, \Delta_i)$, and ending at $\left(r^*, \frac{\pi}{2}\right)$, leading to:

$$\int_{r_M}^{r^*} \left( \frac{\Phi'_{peak}(r)}{v(r)^2} - \frac{1}{r} \right) dr = \int_{\Delta_i}^{\frac{\pi}{2}} \cot(\Delta) d\Delta$$

(A60)

To simplify the resulting expressions, let $F(r)$ be the anti-derivative of $\frac{\Phi'_{\text{peak}}(r)}{v(r)^2}$.

$$F(r) = \int^r \frac{\Phi'_{\text{peak}}(r) dr}{v(r)^2}$$

(A61)

This leads to an integral equation:

$$F(r*) - F(r_M) - \log\left(\frac{r*}{r_M}\right) = -\log(\sin(\Delta_i))$$

(A62)

This equation can be solved for the critical angle $\Delta_i$:

$$\Delta_i = \sin^{-1}\left( \frac{r*}{r_M} e^{F(r_M) - F(r*)} \right)$$

(A63)

The critical angle $\Delta_i$ is a function of the parameters defining the agent behavior, such as $\psi_0$ and $\psi_1$, and the parameters defining the clump, such as the peak occupancy $N$. When $\psi_0$ is very small, trajectories spend much more time under the influence of the potential, and consequently it is much more likely that they are captured by the peak. Thus, for small $\psi_0$, $\Delta_i = \pi/2$. As $N$ increases, the potential becomes stronger, and the values of $\Delta_i$ are increased for all $\psi_0$. When $\psi_0$ is too large $\Delta_i$ goes to $0$ and agents cannot find the peak.
**Appendix Figure 15** contains a plot demonstrating the aforementioned properties of $\Delta_i$.

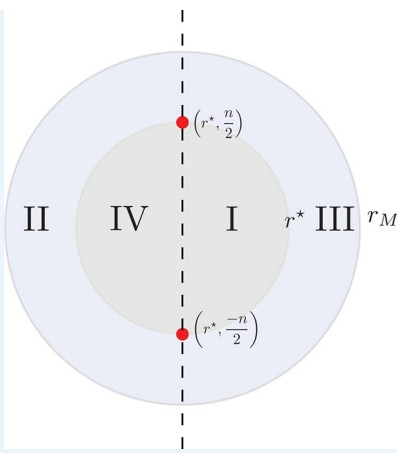

**Appendix Figure 14.** Division of the $(r, \Delta)$ plane into trapping and escaping regions.
Any particle that enters region *I* will eventually reach the zero velocity radius. Any particle that
enters region *II* will escape capture by the peak. Our initial condition will be in the right half
plane, on the circle of width $r_M$. The two red circles are unstable equilibria, each
corresponding to a circular orbit of the peak.

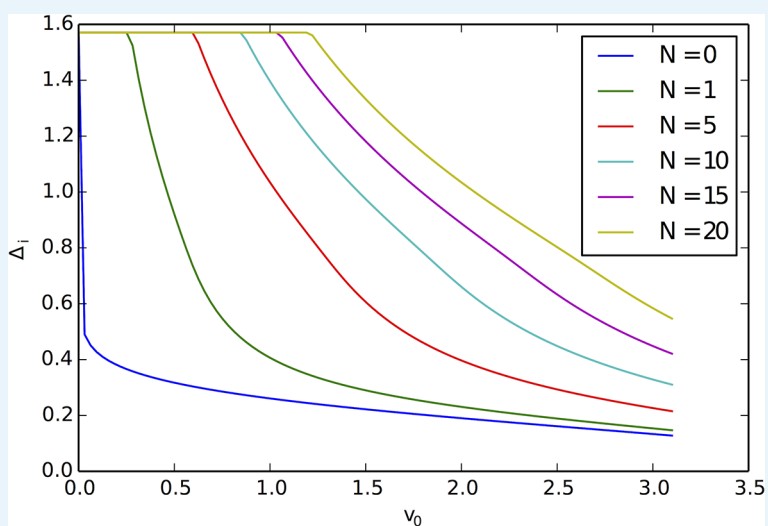

**Appendix Figure 15.** Critical angle $\Delta_i$ for capture of agents by the resource peak, for different
peak occupancy levels *N* and different values of the background velocity $v_0 = \sqrt{\psi_0}$.
Small $v_0$ and large *N* lead to increased $\Delta_i$ and a greater cross-section for capture of agents by
the resource peak.

*Appendix Figure 16* contains a plot of the direction field *Equation A59* and plots of
trajectories that reach $\left(r^*, \pm \frac{\pi}{2}\right)$, demonstrating the trapping of trajectories that enter region $I$,
and providing numerical confirmation of our formula for the critical angle $\Delta_i$.

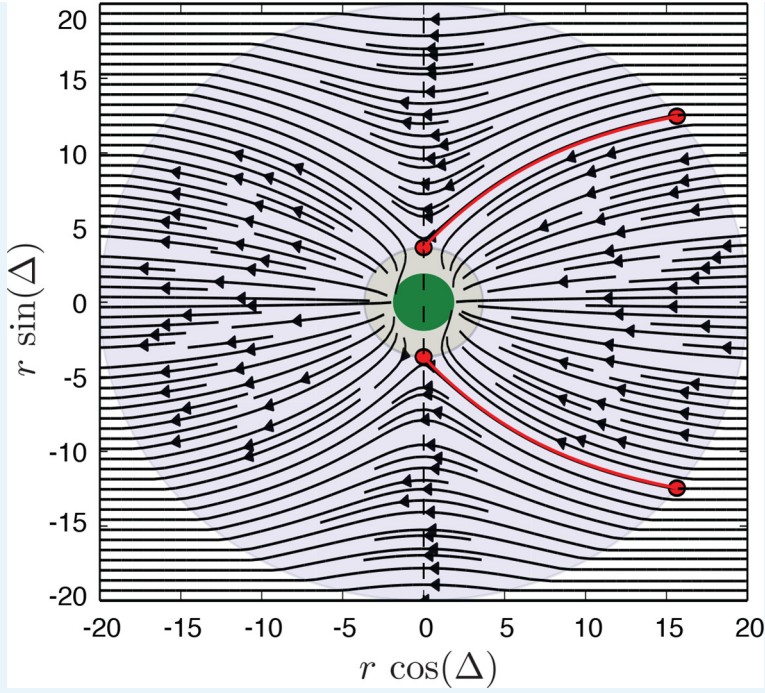

**Appendix Figure 16.** Solutions of the differential equation for $v_0 = 2.0$ and $N = 8$. The green region corresponds to 0 velocity. In the grey region surrounding the green region, the potential is stronger than centrifugal forces, which for $-\frac{\pi}{2} < \Delta < \frac{\pi}{2}$ represents the trapping region. The two red circles correspond to the unstable equilibrium points, and the red trajectories are the trajectories that begin at $(\pm\Delta_i, r_M)$, which reach the equilibrium points and thus represent the boundaries of the set of initial conditions that are captured.

## 6.4 Equation for peak exploration

Using *Equation A63* for $\Delta_i$, we can write an equation for the rate at which the number of agents occupying a resource peak increases.

1. We assume that there is a population of $P$ agents moving in a torus of width $L$, and that $N$ of these agents occupy a resource peak at the origin.

2. The spatial density of agents away from the peak is homogeneous and equal to $\frac{P-N}{L^2}$.

3. The velocity of agents located at $r > r_M$ has magnitude $v_0 = \sqrt{\psi_0}$ and is uniformly distributed in direction.

4. When an agent reaches $r = r_M$, if it has $-\Delta_i < \Delta < \Delta_i$, the agent will be captured by the peak. Otherwise it will escape.

This allows us to calculate the rate of capture of agents on the peak. The flux of agents to the radius $r_M$ and the angle $\Delta$ is equal to $\frac{\rho v_0}{2\pi}$. The flux of agents to a point on the circle with $-\Delta_i < \Delta < \Delta_i$ is equal to $\frac{\rho v_0 \Delta_i}{\pi}$. Then integrating over the circle with radius $r_M$ gives us the total flux to the peak, or the rate of change of the peak occupancy $N$:

$$\frac{dN}{dt} = 2\rho r_M v_0 \Delta_i(N) = \frac{2(P-N)}{L^2} r_M v_0 \Delta_i(N) \tag{A64}$$

## 6.5 Comparison of social versus asocial exploration

We can perform a simple calculation to demonstrate how sociality enhances the rate at which agents occupy a resource peak. In the context of this model, the difference between social

and asocial agents is that the flux of asocial agents to a peak is not enhanced by the presence of agents on a peak. Thus the rate at which the number of asocial agents occupying a peak increases is linear in time. Indeed, if we assume that the total population is large in comparison with the number of agents on the peak, then we can approximate the arrival of asocial agents onto the peak with the following differential equation:

$$\frac{dN}{dt} = 2\rho r_M v_0 \Delta_i(0) \tag{A65}$$

If the peak is unoccupied at time $t = 0$, this equation has solution:

$$N(t) = 2\rho r_M v_0 \Delta_i(0) t \tag{A66}$$

In contrast, the flux of social agents to a peak is enhanced by the presence of other agents on the peak. A similar approximation leads to the equations:

$$\frac{dN}{dt} = 2\rho r_M v_0 \Delta_i(N) \tag{A67}$$

*Appendix Figure 17* contains a plot of the function $\Delta_i(N)$ versus $N$. This plot motivates approximating $\Delta_i(N)$ as a piecewise linear function, linearly increasing from $\Delta_i(0)$ for small $N$ until the value $N_M$ at which $\Delta_i = \frac{\pi}{2}$, at which point the flux becomes a constant function equal to $\pi \rho r_M v_0$. In the initial phase, we approximate the differential equation with:

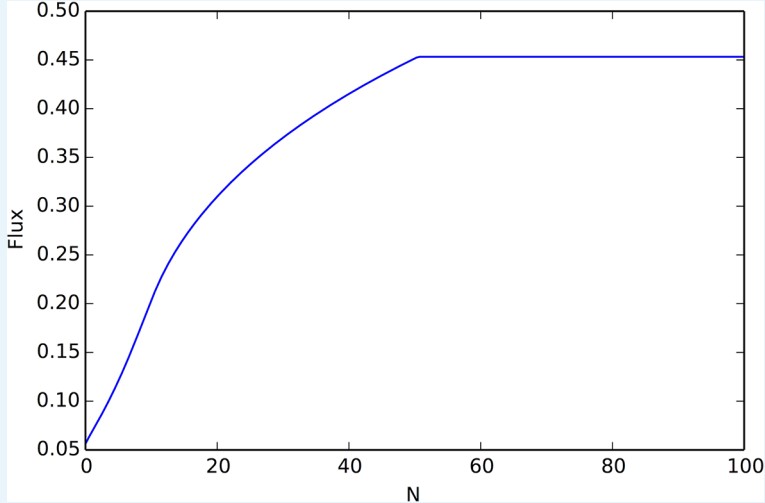

**Appendix Figure 17.** Rate of arrival of social agents onto a resource peak as a function of the number of agents already on the peak.

There are two behavior regimes, the initial regime in which flux grows linearly with *N* (giving rise to exponential growth of the number of individuals on the peak), and a regime where the flux reaches it's maximum value (after which point peak occupancy grows linearly). This figure was calculated using *Equation A67*, with velocity $v_0 = 1.0$, density $\rho = .01$, $K = 25$ interaction neighbors, social potential $\min(N, K)e^{-|r|/7.5}$, and resource peak shape $10e^{-|r^2|/25.0}$.

$$\frac{dN}{dt} = 2\rho r_M v_0 \left( \Delta_i(0) + \frac{N}{N_M} \left( \frac{\pi}{2} - \Delta_i(0) \right) \right) \tag{A68}$$

When the peak is unoccupied at $t = 0$, this has solution:

$$N(t) = \frac{N_M \Delta_i(0)}{\frac{\pi}{2} - \Delta_i(0)} \left( e^{2\rho r_M \nu_0 \frac{t}{N_M} \left( \frac{\pi}{2} - \Delta_i(0) \right)} - 1 \right) \tag{A69}$$

This solution is good up until $N = N_M$, which happens at:

$$t = t_M = \log \left( \frac{\frac{\pi}{2} - \Delta_i(0)}{\Delta_i(0)} + 1 \right) \frac{N_M}{2\rho r_M \nu_0 \left( \frac{\pi}{2} - \Delta_i(0) \right)}. \tag{A70}$$

When $t > t_M$, the solution is:

$$N(t) = N_M + (t - t_M)\pi\rho r_M v_0. \tag{A71}$$

*Appendix Figure 18* compares the function $N(t)$ for social and asocial agents.

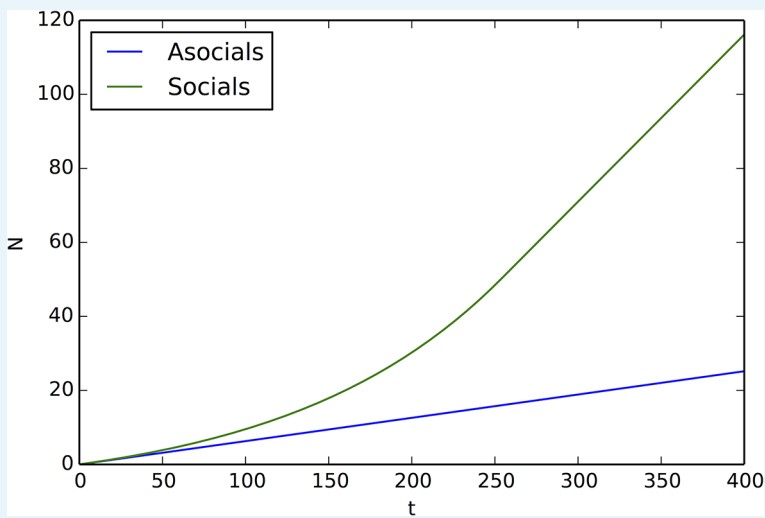

**Appendix Figure 18.** Comparison of peak occupancy as a function of time between social and asocial agents, using the parameters that generated *Appendix Figure 17* and *Equation A66, A69, A71*.
Social agents occupy the peak much more quickly than asocial agents.

# 7 Numerical methods

We used the CVODE subroutine of the SUNDIALS package to numerically solve the agent-based model (*Hindmarsh et al., 2005*). The resulting system of ODEs is stiff, so we utilized the variable order backward-differentiation methods provide by SUNDIALS. We found these implicit methods to be much more efficient than explicit methods for the particular problem that we considered. We also made use of the armadillo linear algebra library (*Sanderson, 2010*), the MATLAB statistics and machine learning toolbox for nearest neighbor searches, and the mex file libraries to interface all of these different tools (*MATLAB, 2015*).

