## [Decision Letter]

Thank you for submitting your work entitled "The evolution of distributed sensing and collective computation in animal populations" for peer review at *eLife*. Your submission has been favorably evaluated by Ian Baldwin (Senior editor), a Reviewing editor, and two reviewers.

The reviewers are enthusiastic about your paper, and we would like to invite you to revise your manuscript according to the suggestions made by the reviewers.

Summary:

This paper considers a model for collective motion in the presence of a time-varying resource environment, in which the characteristics of future generations of the agents evolve according to how well the current generation obtained resources. The model is based on two behavioral rules: a social response rule and an environmental response rule. It is a force-based model, in which the interactions with other agents and the environment are captured in terms of potentials that encode these rules. Parameters that are allowed to evolve include an individual's preferred speed when there is no environmental cue, and the sensitivity of an individual to an environmental cue.

It is found that the parameters evolve to robust evolutionarily stable states, and these are typically places in parameter space near transitions to other collective states. This allows groups to respond quickly to changes in the environment. It is argued that such changes in the collective state resemble a phase transition in physical systems.

The manuscript presents an important and substantial advance in understanding the evolution of collective motion of animals.

*Reviewer #1:*

This paper considers a model for collective motion in the presence of a time-varying resource environment, in which the characteristics of future generations of the agents evolve according to how well the current generation obtained resources. The model is based on two behavioral rules: a social response rule and an environmental response rule. It is a force-based model, in which the interactions with other agents and the environment are captured in terms of potentials that encode these rules. Parameters that are allowed to evolve include an individual's preferred speed when there is no environmental cue, and the sensitivity of an individual to an environmental cue.

It is found that the parameters evolve to robust evolutionarily stable states, and these are typically places in parameter space near transitions to other collective states. This allows groups to respond quickly to changes in the environment. It is argued that such changes in the collective state resemble a phase transition in physical systems.

The paper has a nice combination of results from computations and theory, much of the details of which are described in the Appendix. Overall it is very well written and describes important properties of collective motion systems. I feel that it is a very strong contribution to the literature.

My only "major" comment is the following:

It is common for phase transitions to be characterized in terms of critical exponents that describe how quantities of interest scale near the phase transition. Is it possible to do this for the system in the paper? This would help to strengthen the somewhat vague claim that the changes in the collective state "resemble phase transitions".

Reviewer #2:

In my opinion, the manuscript "The evolution of distributed sensing and collective computation in animal populations" presents an important and substantial advance in understanding the evolution of collective motion of animals and could be published in *eLife*. The authors build up on the relatively well-studied dynamics of non-evolving swarms of interacting particles, introduce important empirically-derived features, and create a comprehensive evolving model which reproduces several common and intuitively-appealing behavioural patterns of such animal groups. In addition to a thorough modeling effort, I especially appreciate the extensive analytic and simplified computational estimates of each important facet of the observed dynamics. Such highly-focused "sub-models" help a reader to verify that the contributions of basic mathematical and biological mechanisms to the rather complex observed phenomenology are understood correctly.

A major issue that naturally comes to one's mind while reading the manuscript is how the results would be affected by adding a realistic feature of a gradual depletion of well-localized resources by swarming consumers. How a rate of such depletion would affect the evolutionary patterns? In the present form, the manuscript is already loaded with new insights, hence, I think, that a qualitative and perhaps even speculative discussion of this topic would suffice.

Reviewer #2 (Minor Comments):

The analogy to the 1-order phase transition based on the density contrast and hysteresis curve could be developed a step further: In traditional first-order phase transitions the hysteresis is caused by the instability of small-size nuclei of an emerging phase. Could something similar be said about the transition between dispersed and cohesive states?

The word "criticality" seems to become too fashionable and abused in many not directly relevant contexts. In its true meaning, a critical state and critical point terminate the line of first-order phase transition rather than sits "close to it". In addition, it is characterized by many unique properties (such as non-analytic behaviour of potentials, scale-free correlations, etc.) which are not mentioned and apparently are irrelevant here. So I would recommend to simply refer to such evolutionary stable state as to one being close to the localized-delocalized transition line.

Same with the word "computation", which, in my understanding, describes some mathematical actions. I think a more appropriate word to describe the collective behaviour would be "collective sensing".

Unless dictated by the journal format requirements, it would be easier on a reader to have the description of the resource dynamics in the Model Development section and to get rid of the Materials and methods one.

The captions to some figures, especially Figure 4 are very difficult to understand. Unless desperately pressed by the size limit, it would definitely help to expand the caption and even split the figure into several ones.

Would it be possible to embed the movies into the main text somewhere near the description of three typical dynamical regimes?

---

## [Author Response]

Reviewer #1:

*[…] It is common for phase transitions to be characterized in terms of critical exponents that describe how quantities of interest scale near the phase transition. Is it possible to do this for the system in the paper? This would help to strengthen the somewhat vague claim that the changes in the collective state "resemble phase transitions".*

We appreciate the reviewer’s point that the analogy between sharp transitions in density/potential energy we observe in collectives, and phase transitions in physical systems could be clarified further. We observe that agents undergo what appears to be a discontinuous change in local density as a function of Ψ, so the observed transition more closely resembles a first order phase transition than a second order phase transition. Because of this, we do not expect the kind of power law scaling near the transition point that is a characteristic of second order transitions, and the search for a critical exponent would not be meaningful. On the other hand, hysteresis is a hallmark first order phase transitions so that Figure 3 and Appendix Figure 9 provide evidence that the analogy to phase transitions is sound. Additionally, we have expanded discussion of the hysteresis behavior present in our model and added additional numerical results (Appendix section 5.4) to further clarify the connection between the behavior of our system and first order phase transitions in physical systems (subheading “Evolved populations are poised near abrupt transitions in collective state”, see also the response to reviewer #2’s comment).

Reviewer #2:

*[…] A major issue that naturally comes to one's mind while reading the manuscript is how the results would be affected by adding a realistic feature of a gradual depletion of well-localized resources by swarming consumers. How a rate of such depletion would affect the evolutionary patterns? In the present form, the manuscript is already loaded with new insights, hence, I think, that a qualitative and perhaps even speculative discussion of this topic would suffice.*

To address the reviewer’s comment, we performed two additional sets of evolutionary simulations under alternative assumptions about resource depletion. In the first alternative scenario, we assumed that individuals deplete the resource very quickly. As one might expect, social cues provide little useful information about the locations of resources when resources are quickly depleted, and as a consequence, the evolutionary trajectories under fast resource depletion resemble those of asocial individuals. Under more moderate levels of resource depletion, however, the evolutionary trajectories shown in Figure 1 of the main text are largely unchanged; social individuals evolve to an evolutionarily stable state in which Ψ0 values are near the transition point in collective state when the resource is at a low value, and individuals cross the transition shown in Figure 3 in regions where the resource value is high. We have added a discussion of these new results to the manuscript (paragraph three, subheading “Evolved populations are poised near abrupt transitions in collective state”).

Reviewer #2 (Minor Comments):

*The analogy to the 1-order phase transition based on the density contrast and hysteresis curve could be developed a step further: In traditional first-order phase transitions the hysteresis is caused by the instability of small-size nuclei of an emerging phase. Could something similar be said about the transition between dispersed and cohesive states?*

As the reviewer mentioned, when Ψ is increased from a low value to a higher value, in the transitional regime, small nuclei of tightly clustered individuals coexist with individuals in the dispersed state. These nuclei eventually become completely unstable at large enough values of Ψ. We have added additional numerical results describing how the rate of nucleus formation depends on Ψ in the hysteresis region and described these in subheading “Evolved populations are poised near abrupt transitions in collective state” of the revised manuscript.

*The word "criticality" seems to become too fashionable and abused in many not directly relevant contexts. In its true meaning, a critical state and critical point terminate the line of first-order phase transition rather than sits "close to it". In addition, it is characterized by many unique properties (such as non-analytic behaviour of potentials, scale-free correlations, etc.) which are not mentioned and apparently are irrelevant here. So I would recommend to simply refer to such evolutionary stable state as to one being close to the localized-delocalized transition line.*

We have removed the term “criticality” throughout the manuscript to avoid confusion about terminology.

*Same with the word "computation", which, in my understanding, describes some mathematical actions. I think a more appropriate word to describe the collective behaviour would be "collective sensing".*

While we agree with the reviewer that the term “computation” is often misused or used casually, in this manuscript we have strived to define precisely what we mean when we use this term (subheading “Changes in collective state allow for rapid collective computation of the resource distribution” and paragraph three, Discussion). One of our motivations for doing this is to help develop a more rigorous definition of “collective computation” as it is a term that is frequently used in the literature on collective behavior.

*Unless dictated by the journal format requirements, it would be easier on a reader to have the description of the resource dynamics in the Model Development section and to get rid of the Materials and methods one.*

As far as we can tell, the Material and methods section is required by *eLife*. If it is allowed, we would be willing to move the details of the resource environment to the Model Development section and omit Materials and methods altogether.

*The captions to some figures, especially Figure 4 are very difficult to understand. Unless desperately pressed by the size limit, it would definitely help to expand the caption and even split the figure into several ones.*

We have divided Figure 4 into two figures and worked to clarify the other figure captions.

*Would it be possible to embed the movies into the main text somewhere near the description of three typical dynamical regimes?*

We have referenced Video 2 in the section of the text that describes dynamical regimes (subheading “Evolved populations are poised near abrupt transitions in collective state”).